# An invasive disease, sylvatic plague, increases fragmentation of black-tailed prairie dog (*Cynomys ludovicianus*) colonies

Krystal M. Keuler[1], Gebbiena M. Bron [1¤], Randall Griebel[2], Katherine L. D. Richgels[1]*

**1** National Wildlife Health Center, U. S. Geological Survey, Madison, WI, United States of America, **2** U. S. Forest Service, Bridger-Teton National Forest, Jackson, WY, United States of America

¤ Current address: University of Wisconsin–Madison, College of Agriculture and Life Sciences, Madison, WI, United States of America
* krichgels@usgs.gov

## Abstract

### Context

A disease can be a source of disturbance, causing population declines or extirpations, altering species interactions, and affecting habitat structure. This is particularly relevant for diseases that affect keystone species or ecosystem engineers, leading to potentially cascading effects on ecosystems.

### Objective

We investigated the invasion of a non-native disease, plague, to a keystone species, prairie dogs, and documented the resulting extent of fragmentation and habitat loss in western grasslands. Specifically, we assessed how the arrival of plague in the Conata Basin, South Dakota, United States, affected the size, shape, and aggregation of prairie dog colonies, an animal species known to be highly susceptible to plague.

### Methods

Colonies in the prairie dog complex were mapped every 1 to 3 years from 1993 to 2015. Plague was first confirmed in 2008 and we compared prairie dog complex and colony characteristics before and after the arrival of plague.

### Results

As expected the colony complex and the patches in colonies became smaller and more fragmented after the arrival of plague; the total area of each colony and the mean area per patch within a colony decreased, the number of patches per colony increased, and mean contiguity of each patch decreased, leading to habitat fragmentation.

### Conclusion

We demonstrate how an emerging infectious disease can act as a source of disturbance to natural systems and lead to potentially permanent alteration of habitat characteristics. While

**Data Availability Statement:** The data underlying the results presented in the study are available at the USGS data repository, Science Base and accessible at https://doi.org/10.5066/P9XON1P4.

**Funding:** The author(s) received no specific funding for this work.

**Competing interests:** The authors have declared that no competing interests exist.

perhaps not traditionally thought of as a source of ecosystem disturbances, in recent years emerging infectious diseases have shown to be able to have large effects on ecosystems if they affect keystone species.

## Introduction

Habitat loss and fragmentation, usually attributed to anthropogenic disturbance or natural disasters, are well documented to affect species diversity and ecosystem processes [1]. An emerging infectious disease that causes a large-scale die-off can be a major disturbance to affected populations, causing population declines or extirpations, altering species interactions, or affecting habitat structure in the short and long term. For example, the introduction of myxomatosis to Great Britain and Australia reduced the abundance of invasive rabbits and allowed native flora and fauna to recover [2], and similarly, the introduction and later eradication of rinderpest altered wildebeest populations leading to widespread changes in the structure of Serengeti grasslands and related ecological processes [3]. In addition, the emergence of sea star wasting disease along the Pacific coasts has led to a major reduction in sea star populations and is expected to cause major changes to intertidal communities [4]. Thus, emerging infectious diseases can have large effects on ecosystem structure and function, especially when a highly contagious and virulent disease is introduced to or invades a naive population and when the primary affected species has a greater effect on the community and ecosystem than would be expected given its numbers or niche (i.e. is a keystone species or ecosystem engineer) [5].

Another example of such a disturbance is plague affecting prairie dogs in the United States. Prairie dogs (*Cynomys* spp.), a keystone species in western US grasslands, northern Mexico and southern Canada [6], currently inhabit approximately 2% of their historical range due to human activities (hunting, poisoning, land-use change) and a re-emerging infectious disease, sylvatic plague [7,8]. These social mammals are ecosystem engineers; they alter the vegetation structure on colonies by removing shrubs and woody plants, increasing the abundance of forbs [9], provide shelter in their burrow networks for small rodents, burrowing owls (*Athene cunicularia*), mountain plovers (*Charadrius montanus*), and other animals [6,10,11], and support diverse predator communities, such as swift foxes (*Vulpes velos*), badgers (*Taxidea taxus*), raptors, and the endangered black-footed ferret (*Mustela nigripes*) [12]. Post plague die-offs, the remaining highly fragmented prairie dog colonies are more susceptible to predation [13], can have reduced gene flow, and are at higher risk of stochastic extinction [14–16]. Consequently, colony fragmentation and habitat loss, in addition to the continued presence of the pathogen on the landscape, threaten the viability of prairie dog populations and the diverse communities associated with them.

Plague, caused by the bacterium *Yersinia pestis*, is a major threat to remaining prairie dog populations. After introduction into San Francisco in 1900, plague spread north and east to its current distribution west of the 100th meridian [17]. While the distribution of plague has been fairly stable, it is continuing to spread slowly east into South Dakota and north into Canada [18,19]. Prairie dogs are highly susceptible to sylvatic plague, and the arrival of *Y. pestis* on a particular prairie dog colony generally leads to population declines [20,21], resulting in a patchwork of affected colonies across the colony complex [22]. Plague epizootics seem to be cyclical, re-emerging on colony complexes 5–15 years after previous plague outbreaks [23]. The mechanism maintaining plague within colony complexes and across the broader landscape is unknown, but potential hypotheses include alternate reservoir hosts, spatial and

temporal rescue effects (e.g. spatial patterns maintain plague on the larger landscape even when absent from local colonies), or low levels of endemicity that are difficult to detect and may affect black-footed ferret survival [24–28].

Previous studies of the landscape patterns of plague in endemic areas (i.e. where plague has been present since ~1950s) have identified climatic and landscape features that affect the patterns of plague occurrence. In particular, colonies with a recent history of plague epizootics are smaller and more isolated than colonies in areas with no recent history of plague [22,29,30]. In addition, colonies that are more connected and in proximity to plague positive colonies are more likely to experience plague die-offs [31]. Lastly, warm winters and wet springs generally support increases in plague activity [32,33], but dry springs can increase flea loads on prairie dogs predisposing them for plague too [34]. However, these studies have focused on areas with long histories of plague caused die-offs, and thus, do not represent the initial effect of this disease upon invasion to a naive colony complex.

Here, we describe the landscape patterns of a colony complex, Conata Basin, South Dakota, that was monitored before and after the initial invasion of plague in 2008. We observed that the invasion of plague led to both loss of occupied grassland and fragmentation of the prairie dog complex. Specifically, we assessed how the arrival of plague in the Conata Basin affected the size, shape, and aggregation of prairie dog colonies. We provide this case study as an example of the role of invading or emerging infectious diseases as a source of disturbance that can lead to habitat loss and fragmentation.

## Methods

### Study area & colony mapping

The study area is considered mixed-grass prairie and consisted of black-tailed prairie dog (*Cynomys ludovicianus*) colonies within the Conata Basin area of Buffalo Gap National Grassland near Badlands National Park in southwestern South Dakota (UTM Zone 13N 677522 East, 4874059 North by 764865 East, 4839493 North; S1 Fig). All of the prairie dog colonies surveyed in the study area were located on public land managed by the U.S. Department of Agriculture, US Forest Service (USFS). Therefore, no permits had to be obtained as the work was conducted under the management authority of the USFS. Interspersed private lands and National Park Service lands were not included, which resulted in un-natural colony edges recognizable by straight lines in Fig 1.

The primary land use of the Conata Basin area is cattle grazing and recreation, however recreational shooting of prairie dogs has not been permitted in most areas since 1998 (Forest

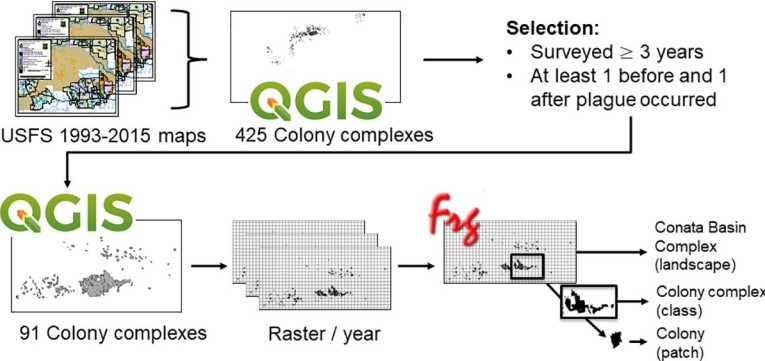

**Fig 1. Composite map of the study area, Conata Basin, South Dakota.**

Service Special Order WRD-98-01). Beginning in 2005, Boundary Management Zones (BMZs) were used in the Buffalo Gap National Grassland in areas ≤ ½ mile adjacent to private lands to reduce unwanted prairie dog colonization onto private land. Within BMZs, the use of rodenticide, vegetation management, and physicl barriers, such as fencing, were allowed [35]. However, only 5 colonies of the 91 selected colonies crossed into private lands and would have potentially been part of the BMZs. Of these, a visual inspection of the spatial data did not show any clear difference in the location of the boundaries by year pre or post 2005. Therefore, it is unlikely to affect trends in fragmentation post 2008. In addition to human land uses and management, prairie dogs were also subject to drought or flood conditions and predation by native predators, including black-footed ferrets that were first reintroduced into the area in 1994 and supplemented during 1996–1999 [36,37].

## Plague arrival

Plague was first detected in South Dakota in 2005, and to be proactive a consortium of management agencies in South Dakota dusted approximately 2,800 ha (8 core colonies representing 27% of colony area and the primary habitat of re-introduced black-footed ferrets) with deltamethrin to reduce flea populations. In 2006 & 2007, 2 of these 8 colonies representing approximately 400 ha were repeatedly dusted. In May 2008, the first severe incidence of plague was observed in Conata Basin. The disease was confirmed through testing of individual deceased prairie dogs on core colonies in concert with large-scale die-offs throughout the prairie dog colonies [38]. Although confirmatory testing was not completed for all potentially affected colonies, there were no other known sources of large-scale mortality co-occurring (such as drought or flooding). Within the full colony complex, 4,000–6,000 hectares of prairie dog colonies were dusted with an insecticide (deltamethrin) annually as of June 2008 to control the spread of disease and preserve the black-footed ferret recovery area [38]. The use of dusting, an effective tool for vector control thereby increasing prairie dog survival [28], will make the estimates of the selected colony fragmentation due to plague invasion conservative, but dusting was necessary to meet conservation objectives. In addition, small-scale experimental plots to test the efficacy of various treatments to prevent plague outbreaks may have occurred in the study colonies. However, generally these experiments have short durations and covered small spatial scales (for example, [21]), thus were unlikely to bias the larger dataset.

## Mapping

The perimeter of active prairie dog colonies present in the study area were mapped by the USFS over the course of 1993 to 2015 every 1 to 3 years. For each year the area was surveyed, all-terrain vehicles were used to record the boundaries of active prairie dog colonies while using a handheld global positioning system (GPS) to record the geographic coordinates every 1 to 5 seconds. Active prairie dog colonies were identified by the visual and audible presence of prairie dogs, and the boundary of these colonies was determined by locating signs of maintenance or recent prairie dog activity. This method was used to estimate colony size in all years except in 1993 and 2006, when the locations of the prairie dog colonies were determined by aerial mapping (for additional methodological details, see [39,40]).

## Spatial analysis

The GPS data were incorporated into vector-based geographic information system (GIS) files, projected to North American Datum 1983, Universal Transverse Mercator zone 13 North (NAD83/UTM Zone 13) by the USFS. Using QGIS 2.14 Essen, any errors in the projected GPS

data were corrected, and a single vector-based GIS file was created for each year the prairie dog colonies were mapped. To generate a unique identification code for colonies through time, the yearly GIS files were merged together to create a composite map of all the colonies surveyed between the years 1993 and 2015. The colonies in the composite map were then buffered by 69m and merged into a single shapefile representing the maximum boundary of the colony from 1993–2015. We used this buffer to identify unique colony complexes because it was the mean distance black-tailed prairie dogs moved in a spatial mark-recapture study (n = 21,000 marked prairie dogs in 58 sites over 7 western states, study described in [21]). The estimate is not intended to capture dispersal and represents a conservative estimate of prairie dog home range size ($(69m)^2$ = 0.48 ha, coterie sizes from South and North Dakota black-tailed prairie dog colonies ranged from 0.5–1.8 ha [41]). We did not take physical barriers, like deep drainages, into account and some colonies could be counted as one, even though prairie dogs do not cross these barriers daily. Through this process, we identified 425 spatially distinct merged colonies. We then used a spatial join to transfer the unique ID of the spatially distinct merged colonies to the yearly shapefiles and refer to them as colony complexes (as depending on the year, there may be one or more spatially distinct colonies with the same unique ID), thus ensuring that patches, referred to as colonies, within each unique colony complex could be tracked through time.

We selected colony complexes for analysis that were repeatedly surveyed throughout 1993–2015. Specifically, colony complexes included in the analysis were sampled at least three times (with a maximum of 12 times), and included at least one measurement before and one measurement after plague arrived. We chose to focus on colony complexes that were measured at least once before and after the arrival of plague as the entire study area was not consistently mapped until 2007. This selection criteria will make our estimates of post-plague fragmentation potentially more conservative. New vector-based GIS files including the 91 selected and unbuffered colony complexes were created for each surveyed year and converted to raster-based GIS files with a resolution of 25 m (the default in ArcGIS 10) to be able to quantify the spatiotemporal changes in FRAGSTATS (Fig 2). The rasterization process can split polygon features into distinct patches if the feature is narrow enough to fill less than ½ of the cell (12.5 $m^2$), and this effect would be present in all by year raster layers.

The raster-based GIS files were analyzed in FRAGSTATS v4.2, where classes were defined as the colony complex ID and patches were spatially distinct colonies within the colony complex. We report variables we expected to change due to plague activity: mean patch (colony) area (AREA_MN), total class (colony complex) area (CA), number of patches (NP), mean proximity index (PROX_MN), mean shape index (SHAPE_MN), and mean contiguity index (CONTIG_MN) of the colonies, at both the colony complex ID (class) and study area (landscape) level, during each year surveyed (Table 1). We chose to use the proximity index (PROX) to evaluate the relative patch isolation, because it considers both the distance of a patch to other patches as well as the size of the patches around it. Therefore, a patch surrounded by multiple other small patches may have the same proximity index as a patch near one large patch, thus providing a measurement of both the degree of patch isolation and the degree of patch fragmentation and offering a more ecological based representation of isolation. To assess possible vulnerability of a patch (edge to interior ratio), we included the mean shape index, which characterizes the mean geometric complexity of each patch by measuring the perimeter-to-area ratio in comparison to a square standard, and the mean contiguity index, which describes the mean contiguity or connectedness of each patch's border configuration.

## Conata Basin/Badlands Praire Dog Complex

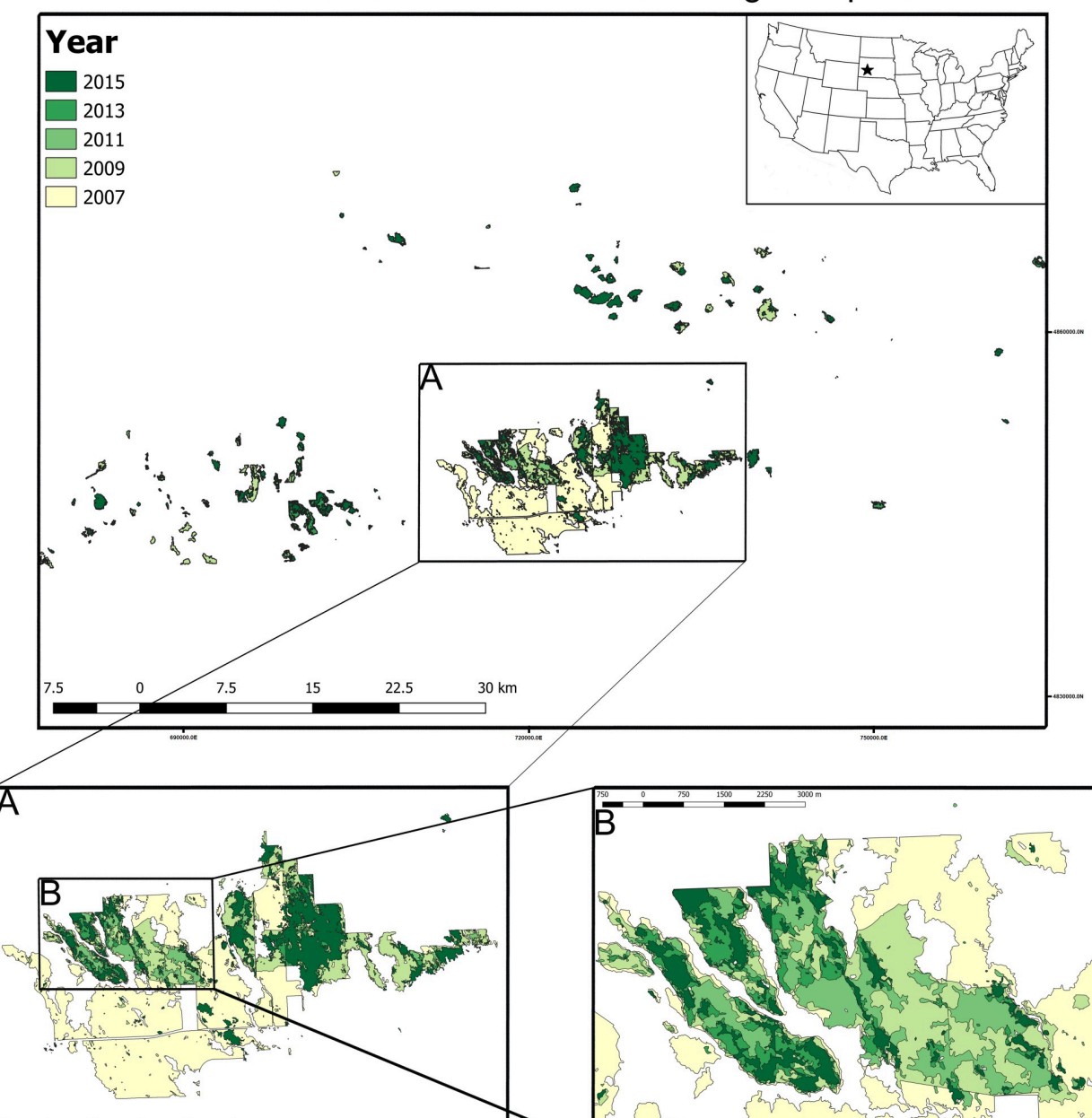

**Fig 2. Workflow from U.S. Forest Service collected shape files to variables used in analysis.** Shapefiles of colony boundaries collected by the USFS from 1993 to 2015 were overlaid and merged in QGIS to generate a composite map, identifying 425 non-overlapping colony complexes. Colony complex ID was added to individual year shapefiles using spatial join. We then applied our selection criteria, and generated year shapefiles with the 91 selected colony complexes. The shapefiles were converted to raster files and analyzed in FRAGSTATS with the colony complex ID as the class feature, generating landscape (Conata Basin complex), class (colony complex) and patch (colony within a complex) variables for statistical analysis in R Statistical Software.

### Data analyses

The spatial characteristics for each colony complex ID were compared before and after plague in Statistical Software R ([43]; see R code in Online Resource 1, the data supporting this

**Table 1. Metrics used in analysis.**

| Metric | Description | Units |
|---|---|---|
| *Total Class Area (CA)* | The total area of a colony complex | Hectares (ha) |
| *Number of Patches (NP)* | The total number of colonies in a colony complex or the entire landscape | None |
| *Mean Patch Area Distribution (AREA_MN)* | The mean area of each colony in a colony complex | Hectares (ha) |
| *Mean Shape Index Distribution (SHAPE_MN)* | A measure of the mean geometric complexity of each colony in a colony complex. It is the perimeter-area ratio adjusted to a square standard. | None SHAPE is always $\geq 1$ If SHAPE = 1, the colony's shape is a square, as SHAPE increases, the geometric complexity of the colony's shape increases. |
| *Mean Contiguity Index Distribution (CONTIG_MN)* | A measure of the mean boundary configuration of the colonies in a colony complex, based on the spatial connectedness of grid cells within the patch. | None $0 \leq CONTIG \leq 1$ If CONTIG = 0, it is a 1 pixel patch, as CONTIG approaches 1, it indicates a more contiguous patch shape. |
| *Mean Proximity Index Distribution (PROX_MN)* | A measure of the mean spatial isolation of colonies to other intra-colony complex or inter-colony complex colonies. | None PROX is always $\geq 0$ If PROX = 0, then the patch has no neighbors of the same patch type within the search radius, as PROX increases, it indicates more neighboring patches of the same type that are closer and/or larger. |

Parameters were computed using FRAGSTATS v4.2.1. If applicable, metrics were computed at both the Landscape and Class level. See FRAGSTATS manual for detailed descriptions and mathematical equations of each metric [42].

manuscript are available at DOI: https://doi.org/10.5066/P9XON1P4). We categorized years before plague as 0 and sequentially numbered the years since the outbreak (e.g. $\leq$ 2008 = 0, 2009 = 1, 2011 = 3) and used this as a continuous explanatory variable in linear mixed effect models using function *lmer* for most variables, and function *glmer* with a Poisson distribution for number of patches, in package lme4 [44]. The models were nested by colony complex ID to assess changes per colony complex over time. Nesting by colony complex ID ensures that model comparisons are looking at changes within colony complexes over time and controls for the fact that not all colony complexes were surveyed each year. Six models were constructed, one for each spatial characteristic: total class area (natural log transformed), number of patches, the mean patch area (natural log transformed), mean shape index, mean contiguity index, and mean proximity index (natural log transformed after patches without nearest neighbors were removed). Model residuals were assessed for normality. For each characteristic, fitted models were compared to random effect only models by Akaike's Information Criterion (AIC) [45]. The model with the lowest AIC was considered the best model and models within 2 AIC were considered equal and the most parsimonious was chosen.

## Results

### Study area and colony mapping

In the study area of approximately 300,000 hectares, 425 unique colony complexes were identified between 1993 and 2015; 91 were included in the analyses. At least 17 and up to 84 colony complexes were included per year.

### Colony size

Prior to the invasion of plague in 2008, the Conata Basin complex expanded (Fig 1, Table 2). In 1993 the number of unique colony complexes selected in the study area was 19, with 221 total number of colonies. By 2007 the total number of colony complexes increased to 66 with 143 total number of colonies. The mean number of colonies (patches) per colony complex also decreased from 11.47±34.20 in 1993 to 2.17±4.60 in 2007, meaning that fragmented colonies merged into larger continuous colonies. During this time of expansion, the total area of the

**Table 2. Summary of fragmentation metrics from FRAGSTATS for the rasters by year of the 91 selected prairie dog colony complex boundaries.** Plague was observed in 2008. Note that not all colony complexes were surveyed each year.

| | Landscape Metrics | | Colony Complex Metrics | | | Colony Metrics | | | | | Within Landscape |
|---|---|---|---|---|---|---|---|---|---|---|---|
| Year | No. of Colony Complexes | Total NP | Median CA (ha) | Mean CA (ha) | Mean NP | Mean AREA_MN (ha) | Mean SHAPE_MN | Mean CONTIG_MN | Median PROX_MN | Mean PROX_MN | PROX_MN |
| 1993 | 19 | 221 | 32.88 (3.31–6016.00) | 372.10 ±1,368.48 | 11.47 ±34.20 | 21.08 ±30.69 | 1.77 ±0.46 | 0.75 ±0.12 | 20.81 (0.00–483.75) | 81.26 ±157.81 | 269.6925 ±745.6769 |
| 1996 | 20 | 147 | 24.19 (4.75–3760.06) | 237.33 ±832.12 | 7.35 ±23.31 | 35.87 ±66.16 | 1.62 ±0.56 | 0.87 ±0.05 | 0.18 (0.00–91.16) | 8.64 ±20.91 | 33.8455 ±152.7645 |
| 1999 | 17 | 128 | 16.69 (0.13–3173.38) | 222.51 ±763.06 | 7.53 ±21.85 | 25.79 ±60.82 | 1.62 ±0.32 | 0.74 ±0.20 | 0.19 (0.00–84.89) | 8.37 ±21.53 | 33.0486 ±148.879 |
| 2002 | 20 | 145 | 15.53 (0.94–4406.44) | 252.17 ±979.26 | 7.25 ±22.40 | 21.38 ±48.37 | 1.62 ±0.48 | 0.74 ±0.14 | 0.17 (0.00–174.17) | 17.39 ±49.38 | 88.2765 ±411.8528 |
| 2004 | 20 | 242 | 14.06 (2.44–6674.13) | 373.59 ±1,484.26 | 12.10 ±34.75 | 15.63 ±25.63 | 1.72 ±0.24 | 0.63 ±0.19 | 4.36 (0.00–892.35) | 81.52 ±217.36 | 522.9193 ±1428.5538 |
| 2005 | 24 | 141 | 27.44 (0.38–9998.69) | 484.67 ±2,030.61 | 5.88 ±16.74 | 29.54 ±35.25 | 1.65 ±0.31 | 0.78 ±0.16 | 0.00 (0.00–1894.77) | 117.07 ±391.78 | 953.0438 ±2514.8213 |
| 2006 | 84 | 178 | 18.72 (0.13–10729.81) | 173.15 ±1,170.73 | 2.12 ±6.52 | 28.89 ±37.09 | 1.61 ±0.31 | 0.83 ±0.14 | 0.00 (0.00–2676.15) | 45.92 ±300.69 | 787.8221 ±2389.5057 |
| 2007 | 66 | 143 | 23.97 (0.25–10731.69) | 216.02 ±1,318.02 | 2.17 ±4.60 | 39.74 ±62.64 | 1.67 ±0.36 | 0.80 ±0.15 | 0.00 (0.00–4854.42) | 87.95 ±597.77 | 1201.6556 ±3307.6457 |
| 2009 | 74 | 231 | 23.28 (1.13–4465.94) | 111.62 ±521.33 | 3.11 ±12.53 | 33.93 ±50.85 | 1.71 ±0.38 | 0.79 ±0.15 | 0.00 (0.00–691.11) | 25.31 ±88.38 | 307.5018 ±1331.6753 |
| 2011 | 78 | 370 | 5.00 (0.06–2582.38) | 57.11 ±293.86 | 4.73 ±18.98 | 12.55 ±26.59 | 1.57 ±0.44 | 0.58 ±0.25 | 0.00 (0.00–480.25) | 16.48 ±61.20 | 202.5207 ±772.1746 |
| 2013 | 69 | 299 | 9.75 (0.25–2007.69) | 58.48 ±244.24 | 4.32 ±17.12 | 16.41 ±26.29 | 1.62 ±0.42 | 0.65 ±0.21 | 0.00 (0.00–382.75) | 21.58 ±69.00 | 151.9447 ±630.4578 |
| 2015 | 76 | 500 | 8.34 (0.19–1884.88) | 50.42 ±218.19 | 6.54 ±26.90 | 10.21 ±18.33 | 1.51 ±0.34 | 0.56 ±0.23 | 1.41 (0.00–214.39) | 20.15 ±46.24 | 91.6928 ±432.912 |

Years when plague was present are shaded light gray. Median and range or mean and standard deviation are provided. NP: number of patches (colonies), CA: class area (colony complex), AREA_MN: mean area of a colony within a colony complex, SHAPE_M: mean shape of colonies within a colony complex, CONTIG_MN: mean contiguous index of colonies in a colony complex, PROX_MN: mean proximity index of a colony with in a colony complex or within the total landscape.

selected colony complexes was largest in 2006 (n = 84) at 14,544 ha. After the arrival of plague, the number of selected colony complexes remained fairly constant, but the total number of colonies increased from 231 in 2009 to 500 in 2015 and the mean number of colonies per colony complex also increased from 3.11±12.53 in 2009 to 6.54±26.90 in 2015. In addition, the mean colony complex size (total class area) decreased, indicating colony complexes were fragmented into smaller colonies (Fig 1, Table 2, S2 Fig). The total area of the selected colony complexes decreased to 3,831 ha by 2015 (n = 76).

Between the years 1993 and 2007, the mean total class area was 291.44 (± standard deviation 106.57) ha, with a mean of 173.13(±43.17) patches throughout the entire landscape. After the invasion of plague, the mean total class area decreased to 69.41(±28.36) during 2009 to 2015. The best model assessing total class area included years post plague invasion (ΔAIC = 41.52, S1 Table) and total class area decreased -0.109 on the natural log scale per year post plague (95% confidence interval (95%CI): -0.141, -0.077). Prior to plague, there was a mean of 6.98 (±3.68) colonies per colony complex, with a mean colony area of 51.63(±28.90) ha. Post plague arrival, the mean number of patches throughout the landscape significantly increased to 352.25 (±115.24) (ANOVA p<0.05), with a mean of 4.67(±1.42) colonies per colony complex. The mean patch area of these colonies decreased to 17.19 (±12.54) ha. The best model assessing

number of colonies per colony complex included years post plague (ΔAIC: 221.98; S1 Table) and estimated that number of colonies per complex increased 0.11 (slope estimate 0.11, 95% CI: 0.096, 0.125) every year after plague arrived. The best model assessing mean colony area included years post plague (ΔAIC = 88.07, S1 Table) and mean colony area decreased with 0.179 on the natural log scale per year after 2008 (slope estimate: -0.179, 95%CI: -0.215, -0.144)

## Colony shape

The mean shape index and mean contiguity index decreased after the arrival of plague compared to the pre-plague values, indicating that colonies were becoming more simplistic in shape, smaller, and less contiguous after plague invasion (Table 2). For both spatial characteristics, the model including the years after plague invasion had the best explanatory ability (ΔAIC: 3.92 and 123.49, respectively; S1 Table). Mean shape index was marginally negatively impacted by plague (estimate: -0.012, 95%CI: -0.022,-0.002); prior to 2008, the mean of the mean shape index of the colonies was 1.69 (±0.10) and after the invasion of plague the mean was reduced to 1.52 (±0.05) (Fig 3). Mean contiguity index was estimated to decrease 0.034 each post-plague year (95% CI: -0.039, -0.028; ΔAIC: 123.49; S1 Table). At the landscape level, the mean of the mean contiguity index before the arrival of plague was 0.70 (±0.11), and after plague arrived it decreased to 0.49 (±0.08).

## Colony aggregation

The mean proximity index of a colony to any other colony prior to the arrival of plague varied greatly between 1993 and 2007 (mean: 558.80±496.31, range 33.0 to 1201.7); in the years just before the arrival of plague (2004–2007) mean proximity index was greater than previous years (866.36 compared to 106.22). Following the invasion of plague, it decreased (mean: 209.96 ±100.75), indicating that colonies were becoming more isolated again (Table 2). Between 1993 and 2007, mean proximity index of a colony in a colony complex to another colony in the

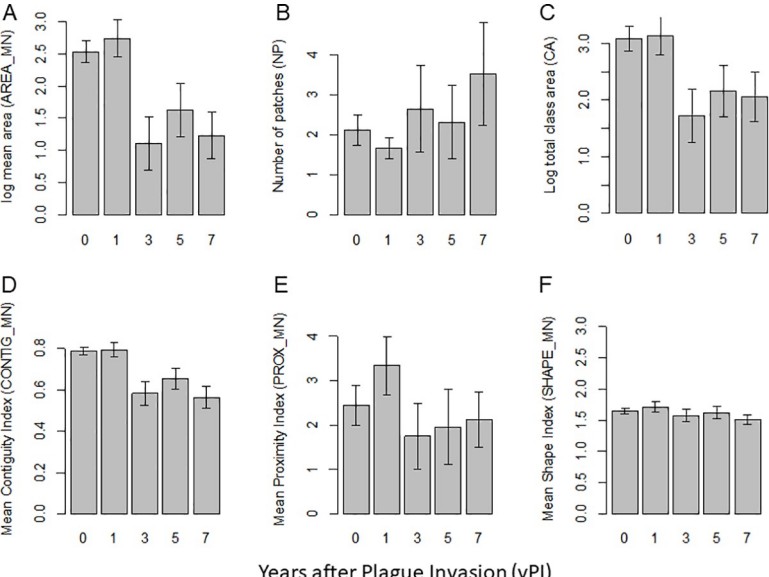

**Fig 3. Colony characteristics change after the arrival of plague.** The largest colony complex was removed from the dataset to create the plot for number of patches. Years post plague invasion (yPI) was a significant predictor (95% CI that did not overlap with zero and included in the best fit model) for all metrics except proximity (PROX_MN). Log = natural log. Error bars represent the standard error.

same colony complex was 56.02 (±41.65). After 2008, this index dropped to 20.88 (±3.65). Years post-plague invasion was not a significant explanatory variable in our within-colony complex mean proximity index model; the best model was the random effects only model (Fig 3; S1 Table).

## Discussion

Spatial characteristics of prairie dog colonies in the Conata Basin prairie dog complex were significantly fragmented by the arrival of plague in 2008. As expected, the total area of each colony complex was reduced by the invasion of plague, which caused a 2/3 decline in prairie dog populations and an estimated 75% decline in the black-footed ferret population [46]. Furthermore, the median total colony complex area decreased (from 23.97 ha in 2007 to a low of 5.00 ha in 2011), signifying that large colonies were becoming less common throughout the complex. In addition, colony complexes were becoming more segmented, as the number of colonies composing each colony complex increased significantly and was positively associated with the number of years since plague had invaded. The mean area of these colonies followed the same pattern as the total colony complex area, meaning that the mean patch size decreased, and smaller patches became more dominant in the years after the arrival of plague. These observations are consistent with previous studies that investigated prairie dog complexes with and without a history of plague occurrence [22,29]. We observed that fragmentation due to plague occurred quickly after invasion to naive complexes and remained as a potentially permanent alteration in this plague-affected habitat [22,29,31], which could be due to slow recovery and recolonization of habitat patches or the presence of enzootic, i.e. low incidence, plague infections that continue to suppress prairie dog populations [28]. Thus, we demonstrate how an invading infectious disease can act as a source of disturbance to natural systems and lead to potentially permanent alteration of habitat characteristics.

Patch complexity is often used as a measure of the strength of edge effects, which can have negative ecological impacts even in the absence of changes in total area [47]. While mean contiguity index and shape index (perimeter/area ratio) were marginally reduced after plague arrived, we suspect this is more related to the reduction in size of colonies leading to slightly simpler perimeters than to any ecologically significant change in patch shape or complexity. Because plague often causes complete eradication of prairie dogs within a particular patch [48], it is unlikely that plague invasion in the short-term would drive changes in the complexity of patch shape.

It was surprising that colony isolation did not increase significantly after plague arrival (as indicated by only marginal significance in proximity index), because Cully et al. [29] observed that the nearest-neighbor distances between colonies in areas with a history of plague were greater than those in areas with no history of plague. Perhaps the most likely explanation is that aggressive plague management via insecticide dust to reduce the flea vector was conducted on 4,000–6,000 hectares per year to maintain prairie dog colonies needed to sustain the resident black-footed ferret population. Unfortunately, because active plague management was on-going and treatment and plague detection histories are not known for each colony, we cannot determine the effect of dusting on the persistence of colonies on the landscape. However, previous research supports that dusting reduces the impacts of plague in prairie dog colonies [49,50]. In addition, plague invasion was well documented within a subset of colony complexes (n = 25), but the dispersal and movement of plague to more isolated colonies was not tracked consistently. Thus, it is possible that plague did not occur on the more isolated colonies in our complex. It is also possible that the invasion of plague was compounded by increased precipitation in 2008 after several years of drought conditions [34]. All of these limitations would bias

our results towards an underestimation of the fragmentation caused by the arrival of plague to Conata Basin. A non-exclusive alternative hypothesis is that insufficient time has passed since the invasion of plague to see the effect of the disease on isolation of colonies within a complex. The sites used by Cully et al. [29] that had increased nearest-neighbor distances had greater than 10 years exposure to plague, without active plague management on the landscape, while our data includes only 7 years since the invasion of plague and a portion under active plague management.

Our study is a conservative measurement of the effects of plague on the Conata Basin prairie dog colonies. To assess the pre- and post-plague spatial characteristics, we only included colony complexes that were measured for at least three different years, with at least one year before the arrival of plague and one year after. This excluded any colony complexes that may have extirpated within the first year of the arrival of plague. If there had been more consistent mapping of colonies in the years before plague arrived, we could have included colonies that were never observed after 2008. Nevertheless, when taken together, our results indicate that plague caused fragmentation of the Conata Basin prairie dog complex, and the fragmentation indicators have so far continued to increase with each year since the onset of plague. It is important to note that the exact arrival date of plague in the Conata Basin is unknown, and it is possible that it was circulating before the first confirmed case.

Because this dataset was collected during management for black-footed ferret recovery, there was a colony complex outlier and a deviation from established data collection methods that could have influenced our results. The prairie dog complex at Conata Basin included one very large colony complex (the black-footed ferret reintroduction location that continues to host a large, self-sustaining population), that at times was a single colony and at other times >100 fragmented colonies. In addition, colony area estimated by the USFS was completed via aerial imagery in 1993 and 2006, which has been documented to overestimate occupied colony area compared to on-the-ground surveys [39]. In both instances, we ran the analyses with these years or colony complex removed and found similar results (analysis results with these removed is provided in the supplemental R code). Excluding these outliers from the dataset significantly reduced the number of colony complexes with at least three time points (the northeastern section of the study area was not surveyed in between 1993 and 2006) and did not significantly change the estimated effects of plague, thus we chose to include them in the final analysis. Lastly, our spatial analysis focused on shape, aggregation, and size of prairie dog colonies. However, other metrics such as burrow density are also important to sustain black-footed ferret populations. Including these metrics was outside the scope of this study, but see [51] for measures of fine-scale changes in burrow densities effects on black-footed ferret populations. Lastly, the data set provided with this analysis is a rasterized subset of colonies in Conata Basin and should not be used as representations of colony acreage occupied by Black-tailed prairie dogs in this area of South Dakota.

Our investigation into the spatial effects of plague in the Conata Basin National Grassland is the first to examine prairie dog colonies before and after the arrival of plague in the area. Because prairie dogs are both a keystone species and an ecosystem engineer, the fragmentation of their colonies caused by plague can have serious ramifications for not only prairie dogs, but the western North American grassland ecosystem as a whole [6,10,27]. The baseline created by this study can be used to evaluate the effectiveness of plague control strategies such as burrow dusting to eliminate fleas and wildlife vaccination programs. The continued surveying of this colony complex, with documentation of treatments and plague detections, will allow for any potential changes to be compared against both the post- and pre-plague data to further evaluate how plague management affects colony size, shape, or aggregation, either by preventing any future colony loss or by allowing the colonies to return to their former configurations [52]

## Conclusion

Overall, we have shown that the invasion of plague has decreased the total area of each colony complex, decreased the mean area per colony within a colony complex, increased the number of colonies per colony complex, as well as marginally decreased mean contiguity of each patch, leading to habitat fragmentation. However, both the mean proximity and shape indexes were only marginally negatively impacted. These findings illustrate the devastating effects of introduced and invading diseases, especially when a keystone species is a highly susceptible host. The disturbances caused by invading diseases can produce significant influences on the landscape and other animals that depend on them. In order to understand the larger ecosystem implications of these types of disturbances, it will be essential to continue monitoring species in both apparently healthy ecosystems and after the invasion of disease.

## Supporting information

**S1 Table. Comparison between the fitted and intercept model for the selected spatial characteristics of each colony.**
(DOCX)

**S1 Fig. A map showing the study area in Buffalo Gap National Grasslands, South Dakota.** The blue shapes represent the buffered colony boundaries used for identifying colonies that belonged to the same complex through the years. Base map and public land boundaries are from the USGS National Map Server (https://viewer.nationalmap.gov/advanced-viewer/).
(TIF)

**S2 Fig. A map of the study area immediately before and after plague invasion.**
(TIF)

**S1 File. R markdown.**
(RMD)

**S2 File. Supplement Keuler et al.** In this file the generalized linear models are ran for the full data set and the reduced dataset (Colony 3 excluded, and aerial surveys [1993, 2006] excluded).
(HTML)

## Acknowledgments

Plague management is a collaborative effort at Conata Basin, and we acknowledge the consortium of agencies that contribute to prairie dog and ferret population management on USFS lands. We thank USFWS and USDA-APHIS, National Park Service, Prairie Wildlife Research, USGS Fort Collins Science Center, and World Wildlife Fund for participating in plague management efforts. In addition, we thank the USFS employees and volunteers who mapped the colonies. We thank T. Rocke, T. Livieri and anonymous reviewers for comments that improved the manuscript. Any use of trade, firm, or product names is for descriptive purposes only and does not imply endorsement by the U.S. Government. The data supporting this manuscript are available at DOI: https://doi.org/10.5066/P9XON1P4.

## Author Contributions

**Conceptualization:** Krystal M. Keuler, Katherine L. D. Richgels.

**Data curation:** Krystal M. Keuler, Randall Griebel, Katherine L. D. Richgels.

**Formal analysis:** Krystal M. Keuler, Gebbiena M. Bron, Katherine L. D. Richgels.

**Investigation:** Randall Griebel.

**Methodology:** Krystal M. Keuler.

**Project administration:** Gebbiena M. Bron, Katherine L. D. Richgels.

**Resources:** Randall Griebel.

**Supervision:** Katherine L. D. Richgels.

**Writing – original draft:** Krystal M. Keuler, Gebbiena M. Bron.

**Writing – review & editing:** Randall Griebel, Katherine L. D. Richgels.

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
