## [Decision Letter · Decision Letter 0]

4 Mar 2020

PONE-D-20-01395

An invasive pathogen, sylvatic plague, increases fragmentation of black-tailed prairie dog (Cynomys ludovicianus) colonies

PLOS ONE

Dear Dr. Richgels,

Thank you for submitting your manuscript to PLOS ONE. After careful consideration, we feel that it has merit but does not fully meet PLOS ONE’s publication criteria as it currently stands. Therefore, we invite you to submit a revised version of the manuscript that addresses the points raised during the review process.

Both reviewers and myself agree that this manuscript is well-written and presents a detailed and interesting contribution to the literature. Both reviewers raise several suggestions for improvement, including more detail on the methods used, consideration of other colony characteristics that could affect plague dynamics in the system, and discussion of enzootic plague. Both reviewers also highlight some nuances with the dataset that come from long-term work in the study region. In particular, reviewer 1 notes that past work on plague mitigation in the region could affect some of the results (and that the authors could acknowledge and discuss these), whereas reviewer 2 emphasizes that some years of data may not be entirely useable given that different methods were used. Lastly, reviewer 1 suggests toning back some statements of novelty, given that some of your findings (e.g., plague-induced prairie dog population destruction) have been well characterized, and thus the authors may wish to instead frame aspects of the work around expanding and building upon prior studies. Also do note that novelty is not a criterion for publication in PLoS ONE. On the topic of publication requirements, the authors do need to report what funding sources were used for the 1993-2015 surveys, as such work undoubtedly entailed various funding efforts.

We would appreciate receiving your revised manuscript by Apr 18 2020 11:59PM. To enhance the reproducibility of your results, we recommend that if applicable you deposit your laboratory protocols in protocols.io, where a protocol can be assigned its own identifier (DOI) such that it can be cited independently in the future. For instructions see: http://journals.plos.org/plosone/s/submission-guidelines#loc-laboratory-protocols

We look forward to receiving your revised manuscript.

Kind regards,

Daniel Becker

Academic Editor

PLOS ONE

Journal Requirements:

2. In your Methods section, please provide additional location information of the study area, including geographic coordinates for the data set if available.

3. In your Methods section, please provide additional information regarding the permits you obtained for the work. Please ensure you have included the full name of the authority that approved the field site access and, if no permits were required, a brief statement explaining why

5.  We note that Figures 1 and 2 in your submission contain [map/satellite] images which may be copyrighted. All PLOS content is published under the Creative Commons Attribution License (CC BY 4.0), which means that the manuscript, images, and Supporting Information files will be freely available online, and any third party is permitted to access, download, copy, distribute, and use these materials in any way, even commercially, with proper attribution. For these reasons, we cannot publish previously copyrighted maps or satellite images created using proprietary data, such as Google software (Google Maps, Street View, and Earth). For more information, see our copyright guidelines: http://journals.plos.org/plosone/s/licenses-and-copyright.

1.     You may seek permission from the original copyright holder of Figure(s) [1 and 2] to publish the content specifically under the CC BY 4.0 license.  

Reviewers' comments:

Reviewer's Responses to Questions

**Comments to the Author**

1. Is the manuscript technically sound, and do the data support the conclusions?

Reviewer #1: Yes

Reviewer #2: Partly

2. Has the statistical analysis been performed appropriately and rigorously? 

Reviewer #1: Yes

Reviewer #2: Yes

3. Have the authors made all data underlying the findings in their manuscript fully available?

Reviewer #1: Yes

Reviewer #2: Yes

4. Is the manuscript presented in an intelligible fashion and written in standard English?

Reviewer #1: Yes

Reviewer #2: Yes

5. Review Comments to the Author

Reviewer #1: This well-written manuscript describes the recent invasion of Yersinia pestis (the causative agent of sylvatic plague) into Conata Basin, South Dakota. The paper summarizes data on prairie dog colony characteristics (e.g., size, shape, aggregation) before and after plague’s invasion. The paper is unique in some respects; prior studies have concentrated (mostly) on areas with historic, endemic plague, and have not included before-after comparisons (but see Eads et al. 2018 for an example from Conata Basin). On the other hand, the paper describes observable, documented phenomena that have been discussed for decades (plague-induced prairie dog population destruction). The manuscript is fascinating and builds upon additional observations/studies that might be discussed in a manner to reduce the tone of novelty.

General comment: The paper concentrates on colony size, shape, and aggregation, which is understandable and appropriate; the underlying data are well suited from such an assessment. That said, additional colony characteristics are important. For example, black-footed ferrets selectively use areas with high densities of active burrow openings (refuge) and prairie dog prey (Biggins et al. 2006, Eads et al. 2011, Livieri and Anderson 2012). These characteristics are extremely important for ferrets, and likely other species. Eads et al. (2018) provide some information on fine-scale changes in burrow/prey density before and after plague’s invasion of Conata Basin. Additional information may be available from those authors and, more generally, groups involved in long-term prairie dog and ferret conservation in the study area.

General comment: The paper concentrates on epizootic plague. Enzootic plague (i.e., transmission between epizootics) is of extreme importance in black-footed ferret and prairie dog conservation (Biggins et al. 2010, Matchett et al. 2010). The authors might briefly mention and describe enzootic plague in the Discussion and emphasize the importance of enzootic plague for wildlife conservation.

General comment: The paper does not mention the decline of black-footed ferrets at Conata Basin. The paper might benefit from some information on the ferret decline (e.g., in the Discussion) given the site is managed (mostly) for ferret conservation. USFS and Prairie Wildlife Research may provide useful information.

Tile: Possibly change “pathogen” to “disease”; the title concentrates on sylvatic plague, not Yersinia pestis.

Keywords: Please change “sciuridae” to Sciuridae.

Line 38: The authors suggest emerging infectious diseases are “not traditionally thought of as a source of disturbance”. The material might be modified; many authors have discussed ecological disturbances and transformations caused by invasive plague (e.g., Antolin et al. 2002, Kotliar et al. 2006, Eads and Biggins 2015).

Line 60: Possibly note that Cynomys occur in Mexico and Canada, as well as the US.

Line 93 and elsewhere, including figures: “Introduction” might be changed to “invasion” to emphasize how plague (probably) arrived at Conata Basin (probably via ecological invasion, though human-assisted introduction is possible).

General comment: In 2007, there was some evidence of sylvatic plague on a colony near Highway 44 (sudden disappearance of prairie dogs documented by USGS-FORT field researchers; plague was not confirmed).

Line 115: The authors might mention weather as another influence on prairie dog populations and colonies.

Line 124: Proactive deltamethrin dusting started in 2005.

Line 124: To my knowledge, 1 of the authors (RG) was intimately involved in plague mitigation and research at the study site. The remaining authors might have been involved peripherally, for a short amount of time. In addition to deltamethrin, several other forms of experimental plague mitigation (vaccines and insecticides) were under use/testing at the study site, especially after plague’s invasion, and those mitigation measures should be discussed, as they might influence results from this study. The authors might contact other researchers intimately involved in plague mitigation and research at Conata Basin, and request help, so the paper fully summarizes all forms of plague mitigation and potential influences on the results. For example, how might the following have affected the results: research on an injectable plague vaccine with prairie dogs and small rodents (USGS-FORT unpublished data); research on an edible vaccine with prairie dogs (Rocke et al. 2017); research on a variety of insecticides (Eads et al. 2019)? How might flea resistance to deltamethrin (Eads et al. 2019) have affected the results?

Line 138: Colonies adjacent to Badlands National Park are cutoff at the park boundary, even though some of the colonies extend into the Park. This phenomenon affects colony size and shape. How might this phenomenon affect the results?

Line 162: Possibly add justification for the use of 25 m resolution.

Line 191: Did the authors consider non-linear forms of “years since outbreak”?

Figure 2: This figure zooms into section B, which includes many colonies treated with deltamethrin and other experimental plague mitigation tools. Consequently, section B concentrates on areas with data that underestimate the effect of plague. The authors might consider adding another figure (even as a supplement) that depicts colony areas in 2007 (on the left) and 2009 (on the right), with no mapping for other years. The change was visually dramatic.

Figure 2: Were Heck Table and areas further to the west included in this study?

Figure 2: Were areas north of the Badlands wall included in this study? If so, the analyses might include a variable for “south” or “north” of the Badlands wall (given plague has most dramatically affected areas south of the wall). The analysis might benefit from a separate assessment of data from south of the wall (alone; this should help to better illustrate the devastating effects of plague, which had not really manifested north of the Badlands wall).

Thank you for the opportunity to review this interesting manuscript. I hope my comments are useful to the authors.

Reviewer #2: Review of PONE-D-20-01395

An invasive pathogen, sylvatic plague, increases fragmentation of black-tailed prairie dog (Cynomys ludovicianus) colonies

Summary: The authors use a prairie dog dataset from Conata Basin, South Dakota to demonstrate how an introduced disease (plague) affects the general landscape characteristics of the prairie dog colonies. The paper is well written and presents quality research that is unique (i.e., before-after effect of plague on prairie dog colonies). The Introduction does an excellent job of describing the background, need for the study, why it is important, and the general objectives. The Methods are well written but need some clarification and justification, primarily in the Spatial Analysis, for the reader to better understand the metrics and how they were measured. My specific comments are below and include censoring the 1993 and 2006 data sets. Figure 1 is excellent in demonstrating the workflow, but could use a little bit more explanation in the text or caption. Figure 2 provides a nice visualization of some of the input data, but is not as vital as Figure 1. Figure 3 provides a good summary of the results. Table 1 is appropriate. Table 2 is relatively dense with information, but the only suggestions I have for improvement would be removal of 1993 and 2006. The Data Analysis is appropriate and well executed. Results are complete and can take a few reads before fully understood. Because the data are rasterized, the reader has to be mindful that the numbers reported are not exactly what was measured on the landscape, but rather a quantitatively accurate generalization. The Discussion and Conclusion appropriately highlight the important findings and place them in the appropriate management context. The authors do a good job of explaining limitations and suggest further research. References may need a few edits. Supplemental Material table is appropriate.

Specific comments:

Line 61: “inhabit only 2% of their historic range…”. Estimates of prairie dog decline vary, thus I suggest a slight change in wording to account for variable estimates, such as “inhabit approximately 2% of their historic range…” See the summary of historic range estimates in Miller et al. 2007 Prairie dogs: an ecological review and current biopolitics. Journal of Wildlife Management 71:2801-2810.

Lines 88-89: “Lastly, warm winters and wet springs…” For an alternate view, see Eads and Biggins 2017 Paltry past precipitation: pre-disposing prairie dogs to plague? Journal of Wildlife Management 81:990-998.

Lines 104-107: “All of the prairie dog colonies surveyed in the study area…” I suggest acknowledging that some prairie dog colonies were contiguous with other federal (i.e., Badlands National Park) or private lands and thus some colony borders analyzed in this study were straight lines (i.e., borders) between land ownership and not necessarily a true characteristic of the colony. Some readers may be confused by some of the straight line borders in Figure 2 and thus suspect that some colonies may be larger or have a different shape than what was analyzed.

Lines 110-114: “Beginning in 2005, Boundary Management Zones (BMZs) were used…” I agree that the consistent BMZ management from 2005-2015 should not be a problem in interpreting fragmentation post-2008. However, prior to 2005 there were often a significant amount of prairie dog colony area in the BMZ area. Did the authors remove the BMZ (i.e., remove the ½ mile) from the colonies prior to 2005? Otherwise the estimates of class area and other metrics (Table 1) presented in Table 2 would be inflated for any colonies prior to 2005.

Lines 125-127: “The use of dusting…” I’m glad the authors acknowledge dusting but it certainly complicates interpretation of the results because it is an effective tool in increasing prairie dog survival at certain levels of plague circulation (Biggins et al. 2010 Vector control improves survival of three species of prairie dogs (Cynomys) in areas considered enzootic for plague. Journal of Vector-Borne and Zoonotic Diseases 10:17-26). Thus the probability of colonies persisting and/or maintaining during plague is possible by dusting. Because dusting is very labor-intensive, it can be difficult to treat entire colonies, potentially creating artificial borders between dusted and non-dusted areas. For instance, in Figure 2, Box B, the large yellow colony in the upper right quadrant has a diagonal line across from SW to NE. That line is very close where dusting occurred (green) and likely influenced the size and shape of that particular colony. Additionally, not every area was continually dusted every year (due to budget) or timing between annual dust applications varied.

Line 131: “…transverse…” I would suggest using a different word as transverse means crossing, often in perpendicular direction. I suggest using “record” or “trace” in this instance as the mappers followed the outline of the colony.

Line 133: “coordinates every second.” For the years 1999-2005 the GPS units recorded a coordinate every 5 seconds (I created the data dictionary and did much of the mapping). Based upon the shape of the colonies in the 1996 coverage, I suspect it was also 5 seconds. Griebel was present 2007-2015 and oversaw all the mapping and can confirm if those years were mapped using 1 or 5 seconds between locations. While this may seem trivial, the interval between locations and the speed traveled can affect the overall shape of a polygon.

Lines 135-137: “This method was used to estimate colony size…” The 1993 coverage was not estimated by GPS ground mapping, but through aerial photography (see Schenbeck and Myhre 1986 Aerial photography for assessment of black-tailed prairie dog management on the Buffalo Gap National Grassland, South Dakota. US Forest Service Technical Report 86-7). I would suggest not using the 1993 coverage because it was derived from aerial photography, the area was open to prairie dog shooting (closure started in 1998, with the exception of a few small areas), and many areas were subject to poisoning, which certainly can impact prairie dog populations. Along those same lines, I would suggest eliminating use of the 2006 coverage as well because it was derived using a different method (aerial photography).

Line 145: “between the years 1993 and 2015”. Again, I would suggest not using the 1993 and 2006 coverage layers because they were collected under a different methodology. Looking at Table 2, there is a marked decrease in some of the metrics between 1993 (aerial photography) and 1996 (GPS ground) that could be an artifact of poisoning, shooting, or methodology. Similarly, the 2006 metrics do not fit the general known pattern (prairie dog colonies increased in size from 1999 to 2007). From my experience on the ground, the colonies did not decrease in size/distribution from 2006-2007 (as Table 2 would suggest and the authors state later in the paper on Line 211).

Lines 145-150: “The colonies in the composite map were buffered…” I understand and support the concept of using a composite map to facilitate the consistent tracking of prairie dog colonies that coalesce/fragment and I agree that 69m is a reasonable daily distance for a prairie dog to move WITHIN a prairie dog colony, but not out from a prairie dog colony. In the Conata Basin, colonies are often separated by barriers that inhibit the daily movements of prairie dogs. The three darkest colonies in Figure 2, Box B are less than 138m from each other, but are separated by deep drainages that are only crossed by prairie dogs during dispersal (certainly not on a daily basis). It’s unclear as to why a buffer is needed from the edge of the colony and some further description/explanation would be appreciated. I’m also having difficulty understanding how the authors had 425 colonies in the composite map. In 2007, there were 40 individual colonies in Box A of Figure 2 (i.e., 40 polygons), which then buffered by 69m results in 9 colonies (i.e., polygons). How did the authors derive 425 individual colonies from a composite map, particularly after buffering by 69m? I may be misunderstanding the methodology.

Lines 155-156: “Selected colonies were sampled…” Please clarify if the sampling was random, stratified, systematic, etc. If I understand this correctly, for each colony, a value was chosen from one coverage year prior to plague and two coverage years after plague. These values were used in the models (later described). Thus the data for colony #28 could be from 2002, 2009, and 2015. And then the data for colony #29 could be from 2005, 2011, and 2013. Why not use all of the data?

Line 157: “…after plagues arrival…” Suggest a change to “…after the arrival of plague…”

Lines 161-162: “…develop raster-based GIS files with a resolution of 25m…” Is this the cell size and then each patch is a contiguous set of cells? I suggest a simple description to help the reader better understand the definition of a patch.

Line 168: I suggest providing a web citation for FRAGSTATS. McGarigal, K., SA Cushman, and E Ene. 2012. FRAGSTATS v4: Spatial Pattern Analysis Program for Categorical and Continuous Maps. Computer software program produced by the authors at the University of Massachusetts, Amherst. Available at the following web site: http://www.umass.edu/landeco/research/fragstats/fragstats.html

Line 204: “…425 unique colonies were identified…” Please clarify how this number was determined per comments above.

Line 209: “The mean number of patches per colony decreased…” The mean number of patches per colony appeared to be stable for 1996, 1999, and 2002. Then a significant increase occurred in 2004, followed by a significant decrease in 2005. Undoubtedly, the colonies were expanding during this time frame and coalescing, but the 2004 data presented in Table 2 appears to be an anomaly? Also, the mean number of patches during the plague years (2009, 2011, 2013, 2015) does increase over time, but not to the same degree as 2002 to 2004.

Line 211: As noted above, I would suggest reconsidering the use of the 2006 data because of the methodology that could account for significant differences. If the 2006 data is used, then it suggests that the total area of prairie dogs decreased in 2007, before the known invasion of plague.

Line 216: Comments on Table 2. In the caption for Table 2, I would suggest adding in a description that this table contains data for the 91 selected colonies, not all of the colonies in the study area. It would be helpful to also describe the data as rasterized so that readers don’t interpret the numbers as actual vector-mapped colony sizes. The definitions of “colony” and “patch” should be clarified for the Table. As I read it, the No. of colonies is the number of composite colonies with a unique ID (out of a total of 425?). In the flowchart of Figure 1, which illustrates the workflow well, the last component visually suggests that patch is a contiguous set of 25m grid cells. If that is the case, then any contiguous prairie dog colony (vector or raster) on the landscape comprises one patch? The Table reports 500 patches in 2015 but the 2015 GIS coverage that I have only has 400 total prairie dog colonies (polygons). Can a contiguous polygon be split in the vector-to-raster process?

Line 298: “…limitation…” Another suggestion could be cascading effects from plague that were difficult to measure, such as increased rainfall (see Figure 11 in Kempema et al. 2015 Colony acreage and distribution of the black-tailed prairie dog in South Dakota, 2012). Drought conditions from 1999-2007 facilitated growth of colonies (e.g., prairie dogs had to seek more forage over a greater area). Above-average rainfall in 2008 (the same year plague invaded) likely also stimulated vegetation growth and the prairie dogs that survived plague were further limited by increased vegetation, thus furthering declines in area, shape, etc. Undoubtedly, plague caused a significant reduction in colony area, shape, and contiguity, but we often speculated that increased rainfall could exacerbate the effects of plague through vegetative growth.

Lines 314-315: “If we had been able to determine…” From my understanding, every colony was revisited during each mapping effort for 2002-2015, thus any colonies that did not appear after 2008 were truly extirpated.

Line 322: “…consisted of one very large colony…” This is confusing related to the 425 colonies reported earlier in the manuscript.

Lines 324-328: “In addition, colony area…” The authors support reasoning for excluding the 2006 coverage from the data set altogether (and the 1993 data set too). I would suggest that keeping the 1993 and 2006 data sets unnecessarily complicates the manuscript and the removal would strengthen the manuscript.

Line 341: “…appearance…” The authors consistently used the word “introduction” in reference to plague and I would suggest using the same word here.

General comments: The GIS coverage data sets for 2007, 2009, 2011, 2013, and 2015 all excluded water sources on the prairie dog colonies (i.e., stock dams) but the prior data sets did not exclude the water sources. On the more recent data sets, this would look like holes within the prairie dog colony, whereas the prior data sets do not have any such holes, although the stock dams were present. It may be helpful to eliminate those stock dams from the prior years, otherwise those would be rasterized cells that would be considered prairie dog colony during the 1993-2006 period and then would not be colony during the 2007-2015 period. That could introduce a small level of bias in the data set.

The References may require some revisions (all Caps in some citations [2], capital letters in some article names [3,4,13,18,21,22,25,32,37,38,39]), although I am not entirely familiar with PLOSOne formatting requirements.

6. PLOS authors have the option to publish the peer review history of their article (what does this mean?). If published, this will include your full peer review and any attached files.

Reviewer #1: No

Reviewer #2: Yes: Travis Livieri

---

## [Author Response · Author response to Decision Letter 0]

16 Apr 2020

Dear Dr. Becker, Dr. Livieri and anonymous reviewer, 

Thank you very much for your thorough review of our manuscript and detailed suggestions for its improvement.

We have tried to address all comments and provided a response below. 

Best Regards,

Dr. Katherine Richgels 

Also on behalf of co-authors 

Journal Requirements:

Response: Done. 

2. In your Methods section, please provide additional location information of the study area, including geographic coordinates for the data set if available.

Response: We have included additional description of the area covered in the methods and will include a shapefile of the colony boundaries selected for our analysis through time with the USGS data release. In addition, a map of the colony area was added as supplemental figure S1. 

3. In your Methods section, please provide additional information regarding the permits you obtained for the work. Please ensure you have included the full name of the authority that approved the field site access and, if no permits were required, a brief statement explaining why

Response: We included a statement explaining why permits were not necessary. “No permits had to be obtained as the work was conducted by the USFS.”

Response: USGS requires all data used for scientific publications be released at the time of publication. The data supporting this manuscript will be made available at DOI: https://doi.org/10.5066/P9XON1P4 once the dataset and manuscript have USGS approval for publication.

5. We note that Figures 1 and 2 in your submission contain [map/satellite] images which may be copyrighted. All PLOS content is published under the Creative Commons Attribution License (CC BY 4.0), which means that the manuscript, images, and Supporting Information files will be freely available online, and any third party is permitted to access, download, copy, distribute, and use these materials in any way, even commercially, with proper attribution. For these reasons, we cannot publish previously copyrighted maps or satellite images created using proprietary data, such as Google software (Google Maps, Street View, and Earth). For more information, see our copyright guidelines: http://journals.plos.org/plosone/s/licenses-and-copyright.

Response: The maps in figure 1 and 2 were created by the authors. QGIS and FragStats are both open source software, their logos are available for use.

Reviewers' comments:

Review Comments to the Author

Reviewer #1: This well-written manuscript describes the recent invasion of Yersinia pestis (the causative agent of sylvatic plague) into Conata Basin, South Dakota. The paper summarizes data on prairie dog colony characteristics (e.g., size, shape, aggregation) before and after plague’s invasion. The paper is unique in some respects; prior studies have concentrated (mostly) on areas with historic, endemic plague, and have not included before-after comparisons (but see Eads et al. 2018 for an example from Conata Basin). On the other hand, the paper describes observable, documented phenomena that have been discussed for decades (plague-induced prairie dog population destruction). The manuscript is fascinating and builds upon additional observations/studies that might be discussed in a manner to reduce the tone of novelty.

Response: Thank you. We have toned down the claims of novelty (not required by PLoS One for publication) throughout.

General comment: The paper concentrates on colony size, shape, and aggregation, which is understandable and appropriate; the underlying data are well suited from such an assessment. That said, additional colony characteristics are important. For example, black-footed ferrets selectively use areas with high densities of active burrow openings (refuge) and prairie dog prey (Biggins et al. 2006, Eads et al. 2011, Livieri and Anderson 2012). These characteristics are extremely important for ferrets, and likely other species. Eads et al. (2018) provide some information on fine-scale changes in burrow/prey density before and after plague’s invasion of Conata Basin. Additional information may be available from those authors and, more generally, groups involved in long-term prairie dog and ferret conservation in the study area. 

Response: We are aware of other efforts to understand prairie dog colony dynamics in the area. However, none of those efforts had the same spatial or temporal scale of this analysis and would require substantial subsetting of included colonies. We have included a statement in the discussion about other metrics that may be important for black-footed ferret recovery (besides size, shape, and aggregation). 

General comment: The paper concentrates on epizootic plague. Enzootic plague (i.e., transmission between epizootics) is of extreme importance in black-footed ferret and prairie dog conservation (Biggins et al. 2010, Matchett et al. 2010). The authors might briefly mention and describe enzootic plague in the Discussion and emphasize the importance of enzootic plague for wildlife conservation.

Response: We have added a sentence about enzootic plague in the discussion. Because we only evaluated 7 years post epidemic, and this was a major fragmenting event (naive populations with invading disease), we felt interpreting the reduced recovery of prairie dog colonies after the outbreak in light of enzootic plague was premature. However, if the time series of colony sizes continues to be collected, it may be possible to evaluate whether there is recovery in between epizootics and thus find evidence for or against the occurrence of enzootic plague in this region. 

General comment: The paper does not mention the decline of black-footed ferrets at Conata Basin. The paper might benefit from some information on the ferret decline (e.g., in the Discussion) given the site is managed (mostly) for ferret conservation. USFS and Prairie Wildlife Research may provide useful information.

Response: We have added some context about management for Black-footed Ferrets throughout. 

Title: Possibly change “pathogen” to “disease”; the title concentrates on sylvatic plague, not Yersinia pestis.

Response: Changed

Keywords: Please change “sciuridae” to Sciuridae.

Response: Thank you for catching that! 

Line 38: The authors suggest emerging infectious diseases are “not traditionally thought of as a source of disturbance”. The material might be modified; many authors have discussed ecological disturbances and transformations caused by invasive plague (e.g., Antolin et al. 2002, Kotliar et al. 2006, Eads and Biggins 2015).

Response: In the case of plague, this has been often discussed. However when thinking about the overall field of ecology, disease is rarely included as a disturbance. We agree with the reviewer that this perception is changing and adjusted our wording. 

Line 60: Possibly note that Cynomys occur in Mexico and Canada, as well as the US.

Response: Southern Canada and northern Mexico were added.

Line 93 and elsewhere, including figures: “Introduction” might be changed to “invasion” to emphasize how plague (probably) arrived at Conata Basin (probably via ecological invasion, though human-assisted introduction is possible).

Response: Thank you for the suggestion. Adjusted throughout the manuscript. 

General comment: In 2007, there was some evidence of sylvatic plague on a colony near Highway 44 (sudden disappearance of prairie dogs documented by USGS-FORT field researchers; plague was not confirmed).

Response: Thank you for sharing that information. We added a comment to the discussion, explaining that Y. pestis might have been present prior to detection.

Line 115: The authors might mention weather as another influence on prairie dog populations and colonies.

Response: Done

Line 124: Proactive deltamethrin dusting started in 2005.

Response: We confirmed that there was some limited dusting on two colonies within the Conata Basin Complex in 2005-2007 for a research project with Dean Biggins. 

Line 124: To my knowledge, 1 of the authors (RG) was intimately involved in plague mitigation and research at the study site. The remaining authors might have been involved peripherally, for a short amount of time. In addition to deltamethrin, several other forms of experimental plague mitigation (vaccines and insecticides) were under use/testing at the study site, especially after plague’s invasion, and those mitigation measures should be discussed, as they might influence results from this study. The authors might contact other researchers intimately involved in plague mitigation and research at Conata Basin, and request help, so the paper fully summarizes all forms of plague mitigation and potential influences on the results. For example, how might the following have affected the results: research on an injectable plague vaccine with prairie dogs and small rodents (USGS-FORT unpublished data); research on an edible vaccine with prairie dogs (Rocke et al. 2017); research on a variety of insecticides (Eads et al. 2019)? How might flea resistance to deltamethrin (Eads et al. 2019) have affected the results?

Response: We acknowledge that the use of experimental treatments to protect prairie dogs, small rodents, and ferrets occurred on small experimental plots that may have been included in our study. However, most of these experiments were conducted in the latter years (for example the sylvatic plague vaccine trials were 2013 - 2016), and on a very small portion of the surveyed colony complexes (the plague vaccine/placebo plots together were 80 ha spread across 2 colonies). The data used in our analysis covers 1993 - 2015 and includes a total colony area of 23,290 ha (though that changes by year). We have added a comment in the methods acknowledging that other experiments were probably on-going in some areas. 

Line 138: Colonies adjacent to Badlands National Park are cutoff at the park boundary, even though some of the colonies extend into the Park. This phenomenon affects colony size and shape. How might this phenomenon affect the results?

Response: Looking more closely at the 91 selected colony complexes, about 15 have artificially constricted boundaries. If colonies grow and constrict in a non-uniform way (e.g. across all edges of the boundary), then we may have underestimated changes in colony shape and complexity for these 15 colonies. However, given this is a smallish subset of the total number of colonies, and the overall strength of the detected patterns, we suspect these artificial boundaries are not overly biasing our results. We have added this detail to the methods.

Line 162: Possibly add justification for the use of 25 m resolution.

Response: Added that it is the default setting in ArcGIS 10.

Line 191: Did the authors consider non-linear forms of “years since outbreak”?

Response: We did not, as the linear models were good fits for the dataset.

Figure 2: This figure zooms into section B, which includes many colonies treated with deltamethrin and other experimental plague mitigation tools. Consequently, section B concentrates on areas with data that underestimate the effect of plague. The authors might consider adding another figure (even as a supplement) that depicts colony areas in 2007 (on the left) and 2009 (on the right), with no mapping for other years. The change was visually dramatic.

Response: We have included the recommended figure in the supplement. 

Figure 2: Were Heck Table and areas further to the west included in this study?

Response: Yes, Heck Table was included. We have added more spatial information as a supplementary figure and in the methods. 

Figure 2: Were areas north of the Badlands wall included in this study? If so, the analyses might include a variable for “south” or “north” of the Badlands wall (given plague has most dramatically affected areas south of the wall). The analysis might benefit from a separate assessment of data from south of the wall (alone; this should help to better illustrate the devastating effects of plague, which had not really manifested north of the Badlands wall).

Response: Yes, sites north of the wall were included. On visual inspection of the yearly shapefiles, there appears to be just as much fragmentation and reduction in colony size North of the wall as South of it. In addition, the North sites were not routinely surveyed until 2006. 

Thank you for the opportunity to review this interesting manuscript. I hope my comments are useful to the authors.

Response: Thank you for your comments! They were exceptionally insightful. 

Reviewer #2: Review of PONE-D-20-01395

An invasive pathogen, sylvatic plague, increases fragmentation of black-tailed prairie dog (Cynomys ludovicianus) colonies

Summary: The authors use a prairie dog dataset from Conata Basin, South Dakota to demonstrate how an introduced disease (plague) affects the general landscape characteristics of the prairie dog colonies. The paper is well written and presents quality research that is unique (i.e., before-after effect of plague on prairie dog colonies). The Introduction does an excellent job of describing the background, need for the study, why it is important, and the general objectives. The Methods are well written but need some clarification and justification, primarily in the Spatial Analysis, for the reader to better understand the metrics and how they were measured. My specific comments are below and include censoring the 1993 and 2006 data sets. 

Response: We have run the analysis with and without 1993 and 2006. There was no change in the significance of the results nor generally the coefficients when removing these years. In addition, removing those years reduced our sample size from 91 to 67. Thus, we have chosen to keep them in the main manuscript but include the reduced models in the supplemental R markdown file.

Figure 1 is excellent in demonstrating the workflow, but could use a little bit more explanation in the text or caption. 

Response: Additional explanation text has been added to the caption. In addition, we have clarified the spatial analysis methods used throughout.

Figure 2 provides a nice visualization of some of the input data, but is not as vital as Figure 1. 

Figure 3 provides a good summary of the results. 

Table 1 is appropriate. 

Table 2 is relatively dense with information, but the only suggestions I have for improvement would be removal of 1993 and 2006. 

The Data Analysis is appropriate and well executed. Results are complete and can take a few reads before fully understood. Because the data are rasterized, the reader has to be mindful that the numbers reported are not exactly what was measured on the landscape, but rather a quantitatively accurate generalization. The Discussion and Conclusion appropriately highlight the important findings and place them in the appropriate management context. The authors do a good job of explaining limitations and suggest further research. References may need a few edits. Supplemental Material table is appropriate.

Response: Thank you for the thorough review of our manuscript. 

Specific comments:

Line 61: “inhabit only 2% of their historic range…”. Estimates of prairie dog decline vary, thus I suggest a slight change in wording to account for variable estimates, such as “inhabit approximately 2% of their historic range…” See the summary of historic range estimates in Miller et al. 2007 Prairie dogs: an ecological review and current biopolitics. Journal of Wildlife Management 71:2801-2810.

Response: Textual change made and reference added.

Lines 88-89: “Lastly, warm winters and wet springs…” For an alternate view, see Eads and Biggins 2017 Paltry past precipitation: predisposing prairie dogs to plague? Journal of Wildlife Management 81:990-998.

Response: Added. 

Lines 104-107: “All of the prairie dog colonies surveyed in the study area…” I suggest acknowledging that some prairie dog colonies were contiguous with other federal (i.e., Badlands National Park) or private lands and thus some colony borders analyzed in this study were straight lines (i.e., borders) between land ownership and not necessarily a true characteristic of the colony. Some readers may be confused by some of the straight line borders in Figure 2 and thus suspect that some colonies may be larger or have a different shape than what was analyzed.

Response: Included and clarified.

Lines 110-114: “Beginning in 2005, Boundary Management Zones (BMZs) were used…” I agree that the consistent BMZ management from 2005-2015 should not be a problem in interpreting fragmentation post-2008. However, prior to 2005 there were often a significant amount of prairie dog colony area in the BMZ area. Did the authors remove the BMZ (i.e., remove the ½ mile) from the colonies prior to 2005? Otherwise the estimates of class area and other metrics (Table 1) presented in Table 2 would be inflated for any colonies prior to 2005.

Response: Only ~5 colonies of the 91 selected crossed into private lands and would have potentially been part of the BMZs. Of these, visual inspection did not show any clear difference in the location of the boundaries by year pre or post 2005.

Lines 125-127: “The use of dusting…” I’m glad the authors acknowledge dusting but it certainly complicates interpretation of the results because it is an effective tool in increasing prairie dog survival at certain levels of plague circulation (Biggins et al. 2010 Vector control improves survival of three species of prairie dogs (Cynomys) in areas considered enzootic for plague. Journal of Vector-Borne and Zoonotic Diseases 10:17-26). Thus the probability of colonies persisting and/or maintaining during plague is possible by dusting. Because dusting is very labor-intensive, it can be difficult to treat entire colonies, potentially creating artificial borders between dusted and non-dusted areas. For instance, in Figure 2, Box B, the large yellow colony in the upper right quadrant has a diagonal line across from SW to NE. That line is very close where dusting occurred (green) and likely influenced the size and shape of that particular colony. Additionally, not every area was continually dusted every year (due to budget) or timing between annual dust applications varied.

Response: The expected effect of dusting was explicitly added to this section. 

Line 131: “…transverse…” I would suggest using a different word as transverse means crossing, often in perpendicular direction. I suggest using “record” or “trace” in this instance as the mappers followed the outline of the colony.

Response: Changed to record.

Line 133: “coordinates every second.” For the years 1999-2005 the GPS units recorded a coordinate every 5 seconds (I created the data dictionary and did much of the mapping). Based upon the shape of the colonies in the 1996 coverage, I suspect it was also 5 seconds. Griebel was present 2007-2015 and oversaw all the mapping and can confirm if those years were mapped using 1 or 5 seconds between locations. While this may seem trivial, the interval between locations and the speed traveled can affect the overall shape of a polygon.

Response: Changed to 1 to 5 seconds. Also, thank you for all your hard work, we feel privileged to work with the dataset! 

Lines 135-137: “This method was used to estimate colony size…” The 1993 coverage was not estimated by GPS ground mapping, but through aerial photography (see Schenbeck and Myhre 1986 Aerial photography for assessment of black-tailed prairie dog management on the Buffalo Gap National Grassland, South Dakota. US Forest Service Technical Report 86-7). I would suggest not using the 1993 coverage because it was derived from aerial photography, the area was open to prairie dog shooting (closure started in 1998, with the exception of a few small areas), and many areas were subject to poisoning, which certainly can impact prairie dog populations. Along those same lines, I would suggest eliminating use of the 2006 coverage as well because it was derived using a different method (aerial photography).

Response: Clarified our recreational shooting statement in line 108-110. Also added, 1993 and Schenbeck reference to the text. We conducted the analysis without 1993 and 2006, please see our response to above comment about why we decided to leave these in the analysis.

Line 145: “between the years 1993 and 2015”. Again, I would suggest not using the 1993 and 2006 coverage layers because they were collected under a different methodology. Looking at Table 2, there is a marked decrease in some of the metrics between 1993 (aerial photography) and 1996 (GPS ground) that could be an artifact of poisoning, shooting, or methodology. Similarly, the 2006 metrics do not fit the general known pattern (prairie dog colonies increased in size from 1999 to 2007). From my experience on the ground, the colonies did not decrease in size/distribution from 2006-2007 (as Table 2 would suggest and the authors state later in the paper on Line 211).

Response: We conducted the analysis without 1993 and 2006 (included in the supplementary R code), see response above. 

Lines 145-150: “The colonies in the composite map were buffered…” I understand and support the concept of using a composite map to facilitate the consistent tracking of prairie dog colonies that coalesce/fragment and I agree that 69m is a reasonable daily distance for a prairie dog to move WITHIN a prairie dog colony, but not out from a prairie dog colony. In the Conata Basin, colonies are often separated by barriers that inhibit the daily movements of prairie dogs. The three darkest colonies in Figure 2, Box B are less than 138m from each other, but are separated by deep drainages that are only crossed by prairie dogs during dispersal (certainly not on a daily basis). It’s unclear as to why a buffer is needed from the edge of the colony and some further description/explanation would be appreciated. I’m also having difficulty understanding how the authors had 425 colonies in the composite map. In 2007, there were 40 individual colonies in Box A of Figure 2 (i.e., 40 polygons), which then buffered by 69m results in 9 colonies (i.e., polygons). How did the authors derive 425 individual colonies from a composite map, particularly after buffering by 69m? I may be misunderstanding the methodology.

Response: We added language to clarify that natural barriers were not accounted for and colonies might have been counted as one even when split by natural barriers, like the deep drainages. The buffering was only used to identify colony complexes through time of the original mapped colony boundaries by year. 425 were the total number of unique colony complexes before our selection criteria were applied. Once we selected colonies with at least 3 surveys, with one before and one after plague arrival we only had 91 unique colony complexes. We have tried to clarify this part of the methods throughout. 

Lines 155-156: “Selected colonies were sampled…” Please clarify if the sampling was random, stratified, systematic, etc. If I understand this correctly, for each colony, a value was chosen from one coverage year prior to plague and two coverage years after plague. These values were used in the models (later described). Thus the data for colony #28 could be from 2002, 2009, and 2015. And then the data for colony #29 could be from 2005, 2011, and 2013. Why not use all of the data?

Response: We use all the years observed for colonies with three or more observations. We clarified this in our methods section. No subsampling was conducted. 

Line 157: “…after plagues arrival…” Suggest a change to “…after the arrival of plague…”

Response: Language was adjusted.

Lines 161-162: “…develop raster-based GIS files with a resolution of 25m…” Is this the cell size and then each patch is a contiguous set of cells? I suggest a simple description to help the reader better understand the definition of a patch.

---

## [Decision Letter · Decision Letter 1]

7 May 2020

PONE-D-20-01395R1

An invasive disease, sylvatic plague, increases fragmentation of black-tailed prairie dog (Cynomys ludovicianus) colonies

PLOS ONE

Dear Dr. Richgels,

Thank you for submitting your manuscript to PLOS ONE. After careful consideration, we feel that it has merit but does not fully meet PLOS ONE’s publication criteria as it currently stands. Therefore, we invite you to submit a revised version of the manuscript that addresses the points raised during the review process.

Both reviewers and myself agree that the revised manuscript has adequately addressed most of the prior concerns. Both reviewers here include a number of more minor comments to improve clarity and transparency of the results.

We would appreciate receiving your revised manuscript by Jun 21 2020 11:59PM. To enhance the reproducibility of your results, we recommend that if applicable you deposit your laboratory protocols in protocols.io, where a protocol can be assigned its own identifier (DOI) such that it can be cited independently in the future. For instructions see: http://journals.plos.org/plosone/s/submission-guidelines#loc-laboratory-protocols

We look forward to receiving your revised manuscript.

Kind regards,

Daniel Becker

Academic Editor

PLOS ONE

Reviewers' comments:

Reviewer's Responses to Questions

**Comments to the Author**

1. If the authors have adequately addressed your comments raised in a previous round of review and you feel that this manuscript is now acceptable for publication, you may indicate that here to bypass the “Comments to the Author” section, enter your conflict of interest statement in the “Confidential to Editor” section, and submit your "Accept" recommendation.

Reviewer #1: (No Response)

Reviewer #2: All comments have been addressed

2. Is the manuscript technically sound, and do the data support the conclusions?

Reviewer #1: Partly

Reviewer #2: Yes

3. Has the statistical analysis been performed appropriately and rigorously? 

Reviewer #1: Yes

Reviewer #2: Yes

4. Have the authors made all data underlying the findings in their manuscript fully available?

Reviewer #1: Yes

Reviewer #2: Yes

5. Is the manuscript presented in an intelligible fashion and written in standard English?

Reviewer #1: Yes

Reviewer #2: Yes

6. Review Comments to the Author

Reviewer #1: This interesting manuscript builds upon existing observations/studies, demonstrating strong influences of plague on prairie dogs. Thank you for the opportunity to review the revision. The authors addressed many of my original comments. Thanks to the authors for their efforts. I hope the following comments are helpful.

Funding and Ethics statements: Operational plague mitigation at Conata Basin has been, and continues to be, highly collaborative. A variety of groups have provided support . It seems the collaborators and funders should be acknowledged (USFS, Prairie Wildlife Research, USFWS, USGS-FORT, World Wildlife Fund, etc.).

Line 65: Although this paper concentrates on a plague epizootic at the Conata Basin, it seems important to emphasize that “enzootic plague” (i.e., plague transmission and host mortality at sub-epizootic levels) also affects prairie dogs and associated species. In fact, enzootic plague may be the primary threat to endangered black-footed ferrets – epizootics are infrequent, and enzootic plague reduces ferret survival by at least 200% during inter-epizootic periods (Matchett et al. 2010). The introduction and discussion should emphasize the importance of enzootic plague so the reader understands the importance of this prominent, albeit less understood portion of plague cycles.

Lines 74-75: The authors suggest, “the arrival of Y. pestis on a particular prairie dog colony generally leads to complete colony collapse”. This statement depends on the definition of “colony collapse”. Generally speaking, for many prairie dog colonies, and large colonies in particular, some prairie dogs remain after an initial epizootic. That said, declines may be defined as “collapse” from the perspective of ferret conservation (if only a few prairie dogs remain, the ferrets will die). The authors cite Pauli et al. (2006) as support. If this sort of statement remains in the paper, the authors should also cite Tripp et al. (2017), who detected functional colony collapse in the presence of epizootics.

Line 76: The authors suggest some colonies are “unaffected”. However, low-level enzootic plague can cause population perturbations that are imperceptible to humans, i.e. unless experiments are conducted (Biggins et al. 2010, Matchett et al. 2010). Seemingly unaffected colonies are probably affected at some level over time, even when epizootics are not observed on those colonies. Enzootic plague deserves greater attention in this manuscript.

Line 77: Please change to “re-emerging in epizootic form”.

Line 81: Please change to “low levels of endemicity that are difficult to detect but, nonetheless, ecologically important” or something similar to emphasize the importance of enzootic plague.

Lines 107 and 389: The authors suggest the work was completed by USFS. However, the number of collaborative groups and agencies is very long (even when considering dusting alone). For example, USFWS provided A LOT of funding for dusting. NPS and USFS work in close collaboration on plague mitigation. Prairie Wildlife Research, USGS-FORT, and World Wildlife Fund contributed to the overall team effort.

Line 115: The authors suggest the BMZ was consistently managed. In some (many) cases, areas within the BMZ were not consistently managed – they were poisoned etc. only when the areas extended to within ½ mile of private lands and (usually) only when private landowners complained.

Line 118: Ferrets were first reintroduced to the area in 1994, and 146 captive-born kits were reintroduced into Conata Basin during 1996-1999 (Livieri 2006, Livieri and Anderson 2012).

Line 122: Please remove “)” at the end of the sentence.

Line 124: Tularemia should NOT be mentioned as a source of large-scale impacts, because tularemia is only known to affect pockets of prairie dogs, not larger colonies (Cherry et al. 2019).

Lines 130-131: Eads et al. (2019) tested a variety of pulicides on multiple colonies, including colonies in the core of Conata Basin, for multiple years.

Lines 141-143: Biologists confirmed, via ground truthing, that some of the aerial-mapped colonies were inactive and therefore should not be included in the current study (because those “colonies” were not really prairie dog colonies – only old burrow mounds remained due to poisoning and other factors).

Lines 151-156: The authors suggest black-tailed prairie dogs use an average area/distance of 69 m per day. They cite data from Rocke et al. (2017). That particular dataset is very large and includes data that were kindly provided by a variety of state, government, non-profit, and Tribal groups. Most of those data are not suited for an evaluation of daily movements, and insufficient for an evaluation of territory sizes (because trapping was of short duration at most sites, within and between years). It’s important to note, data from Rocke et al. (2017) were collected at multiple sites affected by epizootic plague, and plague outbreaks “break” the prairie dog social structure and favor longer movements by animals (Biggins and Eads 2019). When considering more robust datasets, black-tailed prairie dogs move over much shorter distances (e.g., 20-40 m or so, often less than 15 m over monthly periods).

Line 164: The authors commonly use “colony complexes” when referring to single colonies (I think). Complexes were defined by Biggins et al. (1993) and represent collections of colonies in close proximity. If the authors are referring to collections, they might cite Biggins et al. (1993) or provide an alternative definition for their purposes. If the authors are referring to single colonies, then “complexes” should be removed from the manuscript (except when referring to the larger Conata Basin complex).

Line 218: Please explain the cutoff of 2 AIC units. What happens if the authors use the more conservative AICc for small sample sizes?

Line 236 and Results in general: In the discussion, the authors might emphasize that rasterized colony boundaries will undoubtedly overestimate colony sizes (i.e. prairie dogs occupied less acreage than suggested by indices in this paper). This is an important point, because increasing prairie dog acreage may lead to increased complaints from private landowners. USFS and collaborators have reported on colony acreage elsewhere – someone might use this paper to refute pervious reports (which would be inappropriate and unproductive).

Line 294: The authors suggest the local ferret population declined by 1/3. The population declined from >300 down to about 35 (when considering the entire epizootic process).

Line 372: The authors suggest colony maps might be used to evaluate the efficacy of plague mitigation tools. Regarding the current study, it is important to note, after the epizootic most of the remaining colonies had been dusted with 0.05% deltamethrin dust (starting in 2005, in some cases, and continuing through this study). This study shows that 0.05% deltamethrin is a highly effective plague mitigation tool – currently, deltamethrin dust is the best tool (by far) for ferret conservation. The authors should highlight this important point in their manuscript, given their results are highly influenced by dusting (in the absence of dusting, few to no colonies would have remained in the “post-epizootic” period).

Lines 385-387: The authors could cite McDonald et al. (2011) [USGS Scientific Investigations Report 2011-5063]. The supplement suggests research is needed on managed and non-managed habitat to assess plague effects on a variety of species/systems.

Reviewer #2: Thank you for addressing my concerns with the manuscript. The Methods, Spatial Analysis, and additional figures provide a more clear understanding. A few additional minor changes are suggested below:

Line 122: There appears to be an extra ")" after "[34]"

Lines 125-125: As noted by the other reviewer, dusting began in 2005 on 7,000 acres (8 colonies), and repeated in 2006 and 2007 on 1,000 acres (2 colonies). None of the dusting was directly associated with Dean Biggins’ project, but was prophylactic protection and testing equipment/methods for the possible invasion of plague.

Line 346-347: “it is possible that it was circulating well before the first confirmed case.” I would suggest defining what “well before” means. We monitored prairie dog colonies for apparent signs of plague (e.g., burrow activity, colony activity), tested prairie dog carcasses, and measured carnivore seroprevalence. One seropositive coyote was detected in 2005 but no changes in prairie dog activity were documented. Three dead prairie dogs were tested in fall 2007 that were negative, but in the area that plague was confirmed in May 2008. Also, a few fleas collected in fall 2007 contained plague DNA. Thus I would estimate that it was no longer than 1 year before the first confirmed case. I would suggest changing the sentence to read "it is possible that it was circulating up to 1 year before the first confirmed case."

Line 348-349: “Because this dataset was collected during routine management for black-footed ferrets and prairie dog recovery plans”. I suggest changing the wording to “Because this dataset was collected during management of prairie dog colonies for black-footed ferret recovery…” There is not a recovery plan for prairie dogs as they are not a listed species.

7. PLOS authors have the option to publish the peer review history of their article (what does this mean?). If published, this will include your full peer review and any attached files.

Reviewer #1: No

Reviewer #2: Yes: Travis Livieri

---

## [Author Response · Author response to Decision Letter 1]

21 Jun 2020

Dear Dr. Becker, 

Thank you. We addressed comments by the reviewers and additional comments provided by the USGS review process. Please find below our response to the reviewer’s comments, a track changes document and the document with all changes accepted. 

Your Sincerely,

Kathrine L.D. Richgels

 

Reviewers' comments:

Reviewer's Responses to Questions

Comments to the Author

1. If the authors have adequately addressed your comments raised in a previous round of review and you feel that this manuscript is now acceptable for publication, you may indicate that here to bypass the “Comments to the Author” section, enter your conflict of interest statement in the “Confidential to Editor” section, and submit your "Accept" recommendation.

Reviewer #1: (No Response)

Reviewer #2: All comments have been addressed

2. Is the manuscript technically sound, and do the data support the conclusions?

Reviewer #1: Partly

Reviewer #2: Yes

3. Has the statistical analysis been performed appropriately and rigorously?

Reviewer #1: Yes

Reviewer #2: Yes

4. Have the authors made all data underlying the findings in their manuscript fully available?

Reviewer #1: Yes

Reviewer #2: Yes

5. Is the manuscript presented in an intelligible fashion and written in standard English?

Reviewer #1: Yes

Reviewer #2: Yes

6. Review Comments to the Author

Reviewer #1: This interesting manuscript builds upon existing observations/studies, demonstrating strong influences of plague on prairie dogs. Thank you for the opportunity to review the revision. The authors addressed many of my original comments. Thanks to the authors for their efforts. I hope the following comments are helpful.

Funding and Ethics statements: Operational plague mitigation at Conata Basin has been, and continues to be, highly collaborative. A variety of groups have provided support . It seems the collaborators and funders should be acknowledged (USFS, Prairie Wildlife Research, USFWS, USGS-FORT, World Wildlife Fund, etc.).

Reply: Thank you. Author R. Griebel was at USFS Conata Basin from 2007 – 2015, he was unaware of the contribution of these groups before or after his time there. We have added this to the acknowledgments. 

Line 65: Although this paper concentrates on a plague epizootic at the Conata Basin, it seems important to emphasize that “enzootic plague” (i.e., plague transmission and host mortality at sub-epizootic levels) also affects prairie dogs and associated species. In fact, enzootic plague may be the primary threat to endangered black-footed ferrets – epizootics are infrequent, and enzootic plague reduces ferret survival by at least 200% during inter-epizootic periods (Matchett et al. 2010). The introduction and discussion should emphasize the importance of enzootic plague so the reader understands the importance of this prominent, albeit less understood portion of plague cycles.

Reply: The presence of Y. pestis was added to the paragraph as a post-plague die-off problem. While we agree with the reviewer that enzootic plague is important for black-footed ferret survival and long-term species recovery, the manuscript is focused on an epizootic event affecting prairie dogs. We feel that a mention of enzootic plague is appropriate (as updated in the introductory text), but that an extensive review of the topic and its implications would not be appropriate for the context of this particular study.

Lines 74-75: The authors suggest, “the arrival of Y. pestis on a particular prairie dog colony generally leads to complete colony collapse”. This statement depends on the definition of “colony collapse”. Generally speaking, for many prairie dog colonies, and large colonies in particular, some prairie dogs remain after an initial epizootic. That said, declines may be defined as “collapse” from the perspective of ferret conservation (if only a few prairie dogs remain, the ferrets will die). The authors cite Pauli et al. (2006) as support. If this sort of statement remains in the paper, the authors should also cite Tripp et al. (2017), who detected functional colony collapse in the presence of epizootics.

Reply: The wording was rephrased to population declines. 

Line 76: The authors suggest some colonies are “unaffected”. However, low-level enzootic plague can cause population perturbations that are imperceptible to humans, i.e. unless experiments are conducted (Biggins et al. 2010, Matchett et al. 2010). Seemingly unaffected colonies are probably affected at some level over time, even when epizootics are not observed on those colonies. Enzootic plague deserves greater attention in this manuscript.

Reply: Changed wording to remove implication that some colonies are unaffected. 

Line 77: Please change to “re-emerging in epizootic form”.

Reply: Changed to “plague epizootics are cyclical…”

Line 81: Please change to “low levels of endemicity that are difficult to detect but, nonetheless, ecologically important” or something similar to emphasize the importance of enzootic plague.

Reply: Rephrased and added reference to Matchett et al. 2010

Lines 107 and 389: The authors suggest the work was completed by USFS. However, the number of collaborative groups and agencies is very long (even when considering dusting alone). For example, USFWS provided A LOT of funding for dusting. NPS and USFS work in close collaboration on plague mitigation. Prairie Wildlife Research, USGS-FORT, and World Wildlife Fund contributed to the overall team effort.

Reply: Line 107 - we say the land is managed by the USFS, which is correct. However, we have updated the acknowledgement to recognize the consortium of partners who supported black-footed ferret recovery and plague management at Conata Basin.

Line 115: The authors suggest the BMZ was consistently managed. In some (many) cases, areas within the BMZ were not consistently managed – they were poisoned etc. only when the areas extended to within ½ mile of private lands and (usually) only when private landowners complained.

Reply: Changed to better reflect that we don’t think BMZs affected our results because they affected a small number of the total selected colonies (5/91) and there was no signal of this management in the shapefiles.

Line 118: Ferrets were first reintroduced to the area in 1994, and 146 captive-born kits were reintroduced into Conata Basin during 1996-1999 (Livieri 2006, Livieri and Anderson 2012).

Reply: We added the additional ferret reintroduction events and adjusted the citation. 

Line 122: Please remove “)” at the end of the sentence.

Reply: Removed

Line 124: Tularemia should NOT be mentioned as a source of large-scale impacts, because tularemia is only known to affect pockets of prairie dogs, not larger colonies (Cherry et al. 2019).

Reply: Tularemia was removed.

Lines 130-131: Eads et al. (2019) tested a variety of pulicides on multiple colonies, including colonies in the core of Conata Basin, for multiple years.

Reply: Our study period included 1993 - 2015, the study cited by the reviewer started in summer/fall of 2015. Thus, depending on the timing of the colony surveys, Eads et al. 2019 had no or minimal temporal overlap to the data presented here.

Lines 141-143: Biologists confirmed, via ground truthing, that some of the aerial-mapped colonies were inactive and therefore should not be included in the current study (because those “colonies” were not really prairie dog colonies – only old burrow mounds remained due to poisoning and other factors).

Reply: In response to reviewer 2’s comments during the first round of review, we ran the analysis with and without 1993 and 2006. There was no change in the significance of the results nor generally the coefficients when removing these years. In addition, removing those years reduced our sample size from 91 to 67. Thus, we have chosen to keep them in the main manuscript but include the reduced models in the supplemental R markdown file.

Lines 151-156: The authors suggest black-tailed prairie dogs use an average area/distance of 69 m per day. They cite data from Rocke et al. (2017). That particular dataset is very large and includes data that were kindly provided by a variety of state, government, non-profit, and Tribal groups. Most of those data are not suited for an evaluation of daily movements, and insufficient for an evaluation of territory sizes (because trapping was of short duration at most sites, within and between years). It’s important to note, data from Rocke et al. (2017) were collected at multiple sites affected by epizootic plague, and plague outbreaks “break” the prairie dog social structure and favor longer movements by animals (Biggins and Eads 2019). When considering more robust datasets, black-tailed prairie dogs move over much shorter distances (e.g., 20-40 m or so, often less than 15 m over monthly periods).

Reply: The buffering was used to identify colony complexes through time from the shapefiles of mapped colony boundaries by year. Some amount of buffering was necessary to link colonies that fragmented or shifted spatially through time. Published information on prairie dog home ranges was actually quite difficult to find at the time of the analysis (we searched the literature for manuscripts that evaluated home range size and were unsuccessful, leading us to use the data we reference above). However, since we began this project a paper supporting the distance used (from a study that estimated coterie sizes of prairie dog complexes in South and North Dakota) was published. We have added it as support for the distance we chose.

Line 164: The authors commonly use “colony complexes” when referring to single colonies (I think). Complexes were defined by Biggins et al. (1993) and represent collections of colonies in close proximity. If the authors are referring to collections, they might cite Biggins et al. (1993) or provide an alternative definition for their purposes. If the authors are referring to single colonies, then “complexes” should be removed from the manuscript (except when referring to the larger Conata Basin complex).

Reply: The authors understand the confusion and we had similar concerns. However to address concerns by reviewer 2 on how to interpret the represented metrics, we chose to use colony = patch, and colony complex = class. We specified this in the methods: “We then used a spatial join to transfer the unique ID of the spatially distinct merged colonies to the yearly shapefiles and refer to them as colony complexes (as depending on the year, there may be one or more spatially distinct colonies with the same unique ID), thus ensuring that patches, referred to as colonies, within each unique colony complex could be tracked through time.”

Line 218: Please explain the cutoff of 2 AIC units. What happens if the authors use the more conservative AICc for small sample sizes?

Reply: The 2 AIC units follow the guidelines of Burnham and Anderson 2002 for model selection and multi-model inference. Since 567 observations were used for the reported analyses and we only include one parameter in our model the difference between AICc and AIC are a matter of decimals. 

Line 236 and Results in general: In the discussion, the authors might emphasize that rasterized colony boundaries will undoubtedly overestimate colony sizes (i.e. prairie dogs occupied less acreage than suggested by indices in this paper). This is an important point, because increasing prairie dog acreage may lead to increased complaints from private landowners. USFS and collaborators have reported on colony acreage elsewhere – someone might use this paper to refute previous reports (which would be inappropriate and unproductive).

Reply: That is indeed something we would like to avoid. Since we selected for certain colonies, we do not provide an estimate for the full Conata Basin complex and we hope it will not be interpreted this way. In addition, rasterization can both over and underestimate colony size, as a pixel is considered part of the colony if >1/2 of the pixel area is occupied by the shapefile (so depending on colony shape and how it crosses the pixel grid, it can be either over or underestimated). We added a sentence to the discussion clarifying that the numbers reported in this study should not be used as an estimate of colony occupancy across the landscape. 

Line 294: The authors suggest the local ferret population declined by 1/3. The population declined from >300 down to about 35 (when considering the entire epizootic process).

Reply: We adjusted the estimate to a reduction of 75% as described in the black-footed ferret recovery plan. 

Line 372: The authors suggest colony maps might be used to evaluate the efficacy of plague mitigation tools. Regarding the current study, it is important to note, after the epizootic most of the remaining colonies had been dusted with 0.05% deltamethrin dust (starting in 2005, in some cases, and continuing through this study). This study shows that 0.05% deltamethrin is a highly effective plague mitigation tool – currently, deltamethrin dust is the best tool (by far) for ferret conservation. The authors should highlight this important point in their manuscript, given their results are highly influenced by dusting (in the absence of dusting, few to no colonies would have remained in the “post-epizootic” period).

Reply: We know that 4,000-6,000 ha were treated annually after 2008. Unfortunately we are missing two pieces of data that would allow us to evaluate this effect, (1) we do not have a complete record of which colonies were treated and when, and (2) the colonies were not surveyed consistently before 2007, and so it was not possible to estimate the extirpation of colonies due to plague (as opposed to other natural instances, had the time series been more robust). Though we agree with the reviewer that dusting most likely contributed to persistence of prairie dogs on the landscape in Conata Basin, and have made this statement stronger in the discussion. 

Lines 385-387: The authors could cite McDonald et al. (2011) [USGS Scientific Investigations Report 2011-5063]. The supplement suggests research is needed on managed and non-managed habitat to assess plague effects on a variety of species/systems.

Reply: https://pubs.usgs.gov/sir/2011/5063/ Added. 

 

Reviewer #2: Thank you for addressing my concerns with the manuscript. The Methods, Spatial Analysis, and additional figures provide a more clear understanding. A few additional minor changes are suggested below:

Line 122: There appears to be an extra ")" after "[34]"

Reply: Removed

Lines 125-125: As noted by the other reviewer, dusting began in 2005 on 7,000 acres (8 colonies), and repeated in 2006 and 2007 on 1,000 acres (2 colonies). None of the dusting was directly associated with Dean Biggins’ project, but was prophylactic protection and testing equipment/methods for the possible invasion of plague.

Reply: Thank you for informing us! R. Griebel did not begin working at USFS Conata Basin until 2007, and was unaware of these efforts. However, he found documentation of this effort and we have added these details to the methods.

Line 346-347: “it is possible that it was circulating well before the first confirmed case.” I would suggest defining what “well before” means. We monitored prairie dog colonies for apparent signs of plague (e.g., burrow activity, colony activity), tested prairie dog carcasses, and measured carnivore seroprevalence. One seropositive coyote was detected in 2005 but no changes in prairie dog activity were documented. Three dead prairie dogs were tested in fall 2007 that were negative, but in the area that plague was confirmed in May 2008. Also, a few fleas collected in fall 2007 contained plague DNA. Thus I would estimate that it was no longer than 1 year before the first confirmed case. I would suggest changing the sentence to read "it is possible that it was circulating up to 1 year before the first confirmed case." 

Reply: Remove “well” from the sentence.

Line 348-349: “Because this dataset was collected during routine management for black-footed ferrets and prairie dog recovery plans”. I suggest changing the wording to “Because this dataset was collected during management of prairie dog colonies for black-footed ferret recovery…” There is not a recovery plan for prairie dogs as they are not a listed species.

Reply: Reword as suggested.

7. PLOS authors have the option to publish the peer review history of their article (what does this mean?). If published, this will include your full peer review and any attached files.

Do you want your identity to be public for this peer review? For information about this choice, including consent withdrawal, please see our Privacy Policy.

Reviewer #1: No

Reviewer #2: Yes: Travis Livieri

---

## [Editor Report · Decision Letter 2]

25 Jun 2020

An invasive disease, sylvatic plague, increases fragmentation of black-tailed prairie dog (Cynomys ludovicianus) colonies

PONE-D-20-01395R2

Dear Dr. Richgels,

We’re pleased to inform you that your manuscript has been judged scientifically suitable for publication and will be formally accepted for publication once it meets all outstanding technical requirements.

Kind regards,

Daniel Becker

Academic Editor

PLOS ONE

---

## [Editor Report · Acceptance letter]

10 Jul 2020

PONE-D-20-01395R2 

An invasive disease, sylvatic plague, increases fragmentation of black-tailed prairie dog (*Cynomys ludovicianus*) colonies 

Dear Dr. Richgels:

I'm pleased to inform you that your manuscript has been deemed suitable for publication in PLOS ONE. Congratulations! Your manuscript is now with our production department. 

Kind regards, 

on behalf of

Dr. Daniel Becker 

Academic Editor

PLOS ONE